# Pacpaint: a histology-based deep learning model uncovers the extensive intratumor molecular heterogeneity of pancreatic adenocarcinoma

Charlie Saillard [1], Flore Delecourt[2], Benoit Schmauch [1], Olivier Moindrot[1], Magali Svrcek[3], Armelle Bardier-Dupas[4], Jean Francois Emile [5], Mira Ayadi[6], Vinciane Rebours[7], Louis de Mestier[7], Pascal Hammel[8], Cindy Neuzillet[9], Jean Baptiste Bachet [10], Juan Iovanna [11], Nelson Dusetti [11], Yuna Blum [12], Magali Richard[13], Yasmina Kermezli[13], Valerie Paradis[2], Mikhail Zaslavskiy[1], Pierre Courtiol[1], Aurelie Kamoun [1], Remy Nicolle[14] & Jerome Cros [2] ✉

Two tumor (Classical/Basal) and stroma (Inactive/active) subtypes of Pancreatic adenocarcinoma (PDAC) with prognostic and theragnostic implications have been described. These molecular subtypes were defined by RNAseq, a costly technique sensitive to sample quality and cellularity, not used in routine practice. To allow rapid PDAC molecular subtyping and study PDAC heterogeneity, we develop PACpAInt, a multi-step deep learning model. PACpAInt is trained on a multicentric cohort ($n = 202$) and validated on 4 independent cohorts including biopsies (surgical cohorts $n = 148$; 97; 126 / biopsy cohort $n = 25$), all with transcriptomic data ($n = 598$) to predict tumor tissue, tumor cells from stroma, and their transcriptomic molecular subtypes, either at the whole slide or tile level (112 μm squares). PACpAInt correctly predicts tumor subtypes at the whole slide level on surgical and biopsies specimens and independently predicts survival. PACpAInt highlights the presence of a minor aggressive Basal contingent that negatively impacts survival in 39% of RNA-defined classical cases. Tile-level analysis ( > 6 millions) redefines PDAC microheterogeneity showing codependencies in the distribution of tumor and stroma subtypes, and demonstrates that, in addition to the Classical and Basal tumors, there are Hybrid tumors that combine the latter subtypes, and Intermediate tumors that may represent a transition state during PDAC evolution.

Pancreatic ductal adenocarcinoma (PDAC) is predicted to be the second cause of death by cancer in 2030, and its prognosis has seen little improvement in the last decades[1]. PDAC is a highly heterogeneous tumor with preeminent stroma and various histological aspects. Omic studies confirmed its intertumor molecular heterogeneity, possibly one of the main factors explaining the failure of most clinical trials. Two transcriptomic subtypes of tumor cells and stroma, respectively, were described

with major prognostic and theragnostic implications[2–5]. Within the tumor cells, the basal-like subtype is defined by a poorer prognosis linked to early metastases and Folfirinox resistance compared to the classical subtype characterized by a progenitor epithelial phenotype with altered metabolism[6]. Within the stroma, the activated stroma is enriched in disorganized pro-tumor cancer-associated fibroblasts with little extracellular matrix, while the inactive stroma is characterized by abundant and dense collagen secreted by more quiescent myofibroblasts. As of today, these subtypes can only be defined by RNA profiling. Tools were proposed to phenotype PDAC, either with a binary classification of tumor cells (PurIST basal-like/classical) or with four semi-quantitative transcriptomic signatures of tumor (Basal-like, Classical) and stroma (Activated, Inactivated) subtypes[5,7]. The latter approach has the advantage of acknowledging the possibility of intratumor heterogeneity as tumors are "scored" for each signature. Yet, these approaches are limited by the quantity and quality of the samples (formalin fixation and low cellularity like in biopsies) as well as by the analytical delay and trans-platform reproducibility that restrict its application in clinical trials and in routine care. In addition, tumors may harbor a mixture of several subtypes complicating their interpretation using bulk transcriptomic approaches, thereby limiting their clinical value[8]. A recent study suggested that tumor cell architecture (i.e., formation of glands) correlated partially with tumor cell transcriptomic subtypes in primary resected tumors[9]. This approach requires highly trained pathologists and the manual analysis of the whole tumor to fully assess the tumor subtype and its possible heterogeneity. Artificial intelligence was proven to be a valuable tool to predict molecular alterations or phenotypes from histological slides, potentially unlocking advanced diagnosis for all[10–13].

Here, we propose PACpAInt, a multi-level artificial intelligence-based tool using deep-learning models to determine PDAC molecular subtypes (tumor and stroma) from routine histological preparation (Hematoxylin-eosin-Safran (HES staining)), at a resolution enabling to decipher intratumor heterogeneity on a massive scale and providing, in addition, spatial information of the different cell types and their molecular phenotype (Fig. 1a).

## Results

### PACpAInt predicts neoplastic areas and tumor molecular subtypes at the whole-slide level

The models were trained and validated on multiple cohorts using 1796 slides (598 patients) with the corresponding transcriptome (cohorts are described in Fig. 1b). Deep-learning models were trained on a discovery cohort (DISC cohort) composed of 424 whole-slide histological images from 202 PDAC resected in three centers (mean number of slides/case = 2). These models were externally validated in (i) three cohorts of resected PDAC: two from a fourth center (distinct from the training centers) (BJN_U (*n* = 148), BJN_M (*n* = 97)) and the public TCGA_PAAD cohort (*n* = 126) and (ii) one cohort of liver PDAC metastases obtained by fine-needle biopsies (Liver_FNB, *n* = 25) also coming from the fourth center, distinct from the training centers. During the surgical specimen routine examination, the pathologist samples multiple tumor areas leading to several tissue blocks for one patient. For the discovery cohort and two validation cohorts (BJN_U and TCGA), for each patient, the transcriptome was obtained from a small tissue core taken in a block that may or may not correspond to the HES slides analyzed by the models (i.e., "spatially unmatched" cohorts). In contrast, for the BJN_M and Liver_FNB cohorts, the same block was used to generate the HES slide and to extract the RNA. In addition, instead of using a small tissue core, the whole tumor area of the block was microdissected to have a perfect match between the HES and the transcriptome (i.e., "spatially matched" cohorts).

For all the models, the slides were split into smaller images, called "tiles", of 112 × 112 μm (224 × 224 pixels). Models were trained using labels either defined at the tile-level (to detect tumor areas) or at the whole-slide level (to predict tumor subtype). A first model (PACpAInt-Neo) based on two pathologists' annotations was developed to predict neoplastic areas. PACpAInt-Neo successfully detected neoplastic regions when applied to two independent validation cohorts (AUC: BJN_U = 0.99, TCGA = 0.98) (Suppl Fig. 1a). This model allowed the quick identification and visualization of tumor areas, even if they were not contiguous, detecting with high accuracy tumor cell and stroma areas from digestive wall components, lymph nodes, etc. (Suppl Fig. 1b, c). The second model (PACpAInt-B/C) was trained on the DISC cohort using only the areas predicted to be neoplastic to determine the basal-like/classical (B/C) subtypes defined by the PurIST-RNA classifier (Fig. 2a). Despite the high histological diversity of PDAC, PACpAInt-B/C identified a set of morphological features specific to basal-like and classical subtypes (Fig. 2b and Suppl Fig. 2b). For the slide level prediction of the molecular subtype, PACpAInt-B/C assigned a score to a tile subset and aggregated all the tile scores into a single prediction for the slide, i.e., basal-like or classical. PACpAInt-B/C successfully predicted basal-like/classical RNA subtypes in 2 spatially unmatched validation cohorts, with AUCs of 0.86 [0.79−0.94] and 0.81 [0.71−0.90] in BJN_U and TCGA cohorts respectively (Fig. 2c and Suppl Fig. 2a). Comparable performance was achieved on a third validation cohort with spatially matched histological and molecular areas (BJN_M, AUC = 0.83 [0.73−0.93]) (Fig. 2d). To ensure that PACpAInt-B/C was not only recapitulating the histological differentiation, we assessed the performance of the model in well/intermediate and poorly differentiated tumors separately in the three surgical validation cohorts (Supplemental Table 1). AUC ranged from 0.71 to 0.90, far from the random guess, confirming that differentiation and molecular subtypes are not surrogates of one another.

### PACpAInt highlights PDAC intratumor macroheterogeneity and its prognostic impact

Given the previously described intratumor heterogeneity that may blur the transcriptomic labels, we restricted the analysis to the 50% of cases that had the clearest, unambiguous transcriptomic subtype (i.e., cases whose transcriptomic profile suggested that they were "pure", composed of only basal-like or classical cells). In these cases, the performance of the model improved substantially (AUC of 0.91 [0.84−0.98] and 0.88 [0.79−0.98] in the BJN_U and TCGA cohorts, respectively (Fig. 2c and Suppl Fig. 2a)[8,14]. This was particularly significant within the spatially matched validation cohort BJN_M (AUC = 0.95 [0.90−1.0]), highlighting the limitations of the RNA-based binary classification in highly heterogeneous tumors (Fig. 2d) as previously reported, rather than a true increase in the model performance that must be appreciated on the whole cohort[14,15]. Because most patients are diagnosed at the metastatic stage on liver biopsies, we also validated PACpAInt-B/C on 25 fine-needle liver biopsies with matched RNAseq data (Liver_FNB cohort). Performance remained as good as for surgical specimens (AUC = 0.85 [0.69−1.0]), and similarly improved in cases with a clear homogeneous molecular subtype (AUC = 0.92 [0.77−1.0] (Fig. 2e). To assess the robustness of PACpAInt-B/C, we performed on the Liver_FNB cohort, a subsampling of the tiles to mimic tumor-poor biopsies. Using 75% of the tumor tiles, the AUC was similar (AUC = 0.85 [0.68−0.96]) and decreased but remained good when using only 50 or 25% of the tumor tiles to predict the molecular subtypes (AUC = 0.82 [0.62−0.96] and 0.82 [0.61−0.96], respectively). We included PACpAInt-B/C in a multivariate survival analysis in the pooled BJN_U and BJN_M cohorts. PACpAInt-B/C predictions had a strong independent prognostic value on both OS (HR = 1.37 [1.16−1.62] *p* < 0.001) and DFS (HR = 1.27 [1.08−1.49] *p* = 0.003), contrary to the PurIST-RNA classification which was not associated with OS (Fig. 2f, g and Suppl Fig. 2c, d and Supplemental Tables 2, 3).

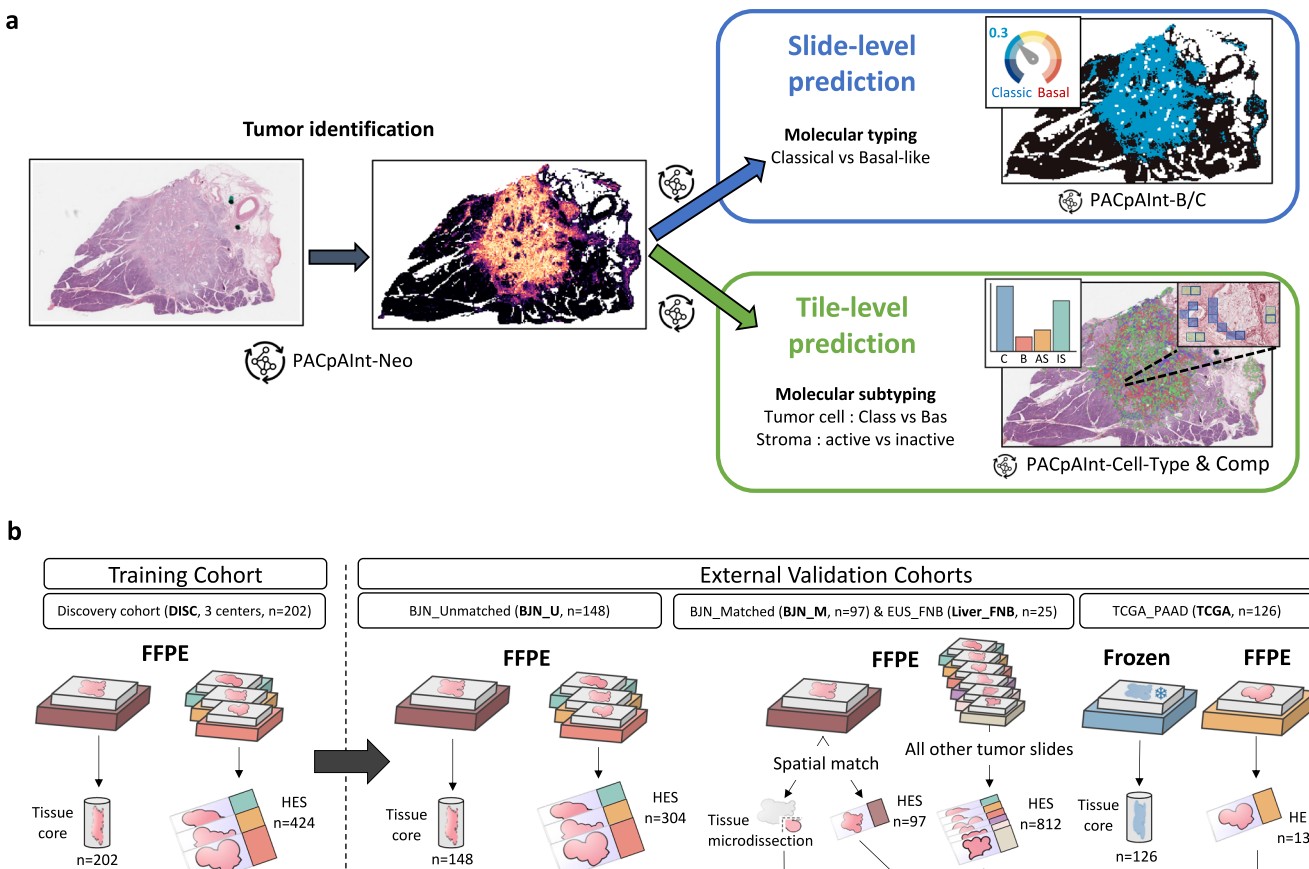

**Fig. 1 | PACpAInt approach for identification of PDAC molecular tumor subtypes.** **a** Simplified workflow of the study: a first model is applied to find the tumor area (tumor cells and stroma) (PACpAInt-Neo) followed by a second model predicting either the global tumor cell molecular type (classical vs basal-like) at the slide level (PACpAInt-B/C) or predicting at the tile level (small square 112 µm wide) the nature of the cells (tumor or stroma - PACpAInt-cell type) and their molecular subtype (classical vs basal-like for tumor cells and active vs inactive for stroma) (PACpAInt-Comp), **b** Description of the cohorts. Discovery cohort (DISC) was composed of 202 patients (surgical specimens) from three centers. A tissue carrot (diameter 600 µm) was taken from a block for RNA profiling. HES slides (at least 2/ tumor) were digitized for PACpAInt analysis. In most cases, the tissue carrot and the HES did not come from the same block. The workflow was similar in the first validation cohort BJN_U unmatched (surgical specimens). For the two next validation cohorts (BJN-M matched (surgical specimens) and EUS_Liver (liver metastases, fine-needle biopsies)), the same block was used for RNA extraction after microdissection of the neoplastic area and to generate the HES slide that was digitized and analyzed with PACpAInt. In addition, in the BJN_M matched cohort, all the remaining tumor slides were also digitized for PACpAInt analysis. Finally, in the TCGA_PAAD validation cohort (surgical specimens), in contrast to all the other cohorts, the RNA was extracted from frozen material, not formalin-fixed paraffin-embedded. Similarly to the discovery cohort, the tissue analyzed by RNAseq was not spatially matched with the digitized slides.

It has been previously shown that tumor cells may harbor distinct morphology from slide to slide within a case[9]. This is particularly meaningful in tumors of the classical subtype, which could be expected to harbor small basal-like areas that could impact patient prognosis. In order to assess the impact of putative minor basal-like areas, we selected in the BJN_M validation cohort the 77 cases (of 97) predicted classical by PurIST-RNA. PACpAInt-B/C was then run on all tumor slides (mean number of slides per case = 9) and we compared the predictions across slides (Fig. 3). Thirty cases (39%) had at least one slide predicted as basal-like, suggesting an important morphological and molecular heterogeneity within those tumors at the scale of the entire lesion. DFS and OS of these heterogeneous cases were shorter (median survival of 15 vs 35 months, $p = 0.08$ and of 36 vs 64 months, $p = 0.002$, respectively), highlighting the clinical impact of tumor heterogeneity (Fig. 3).

### PACpAInt can decipher intratumor microheterogeneity
Regarding the RNA-based tumor characterization, these results prompted us to switch from a dichotomic label (i.e., PurIST-RNA) to

continuous multiparametric labels based on the signatures from Puleo et al. (Suppl Fig. 3a)[5]. Using this method, each tumor was defined by four continuous components, two describing tumor cells (classical and basal-like components) and two describing the stroma (active and inactive). This approach acknowledges that a case can be "pure" with a classical phenotype and an inactive stroma, for instance (Suppl Fig. 3a case 1) or "complex" with the coexistence of both tumor phenotypes (Suppl Fig. 3a case 2), likely reflecting more accurately tumor biology. A new model was trained on the DISC cohort to predict these tumor cell/stroma components on whole histological slides (PACpAInt-Comp). The correlation of PACpAInt-Comp with RNA tumor and stroma components was highly significant in the BJN_U validation cohort and substantially improved in the spatially matched validation cohort BJN_M (Suppl Fig. 3b).

To better study the intratumor microheterogeneity, we then developed a model to predict if a tile was predominantly composed of tumor cells or stroma (PACpAInt-Cell Type) (Fig. 4a). This model was trained on tumor cells and stroma annotations made by an expert pathologist. The model reached an AUC of 0.99 in the two validation

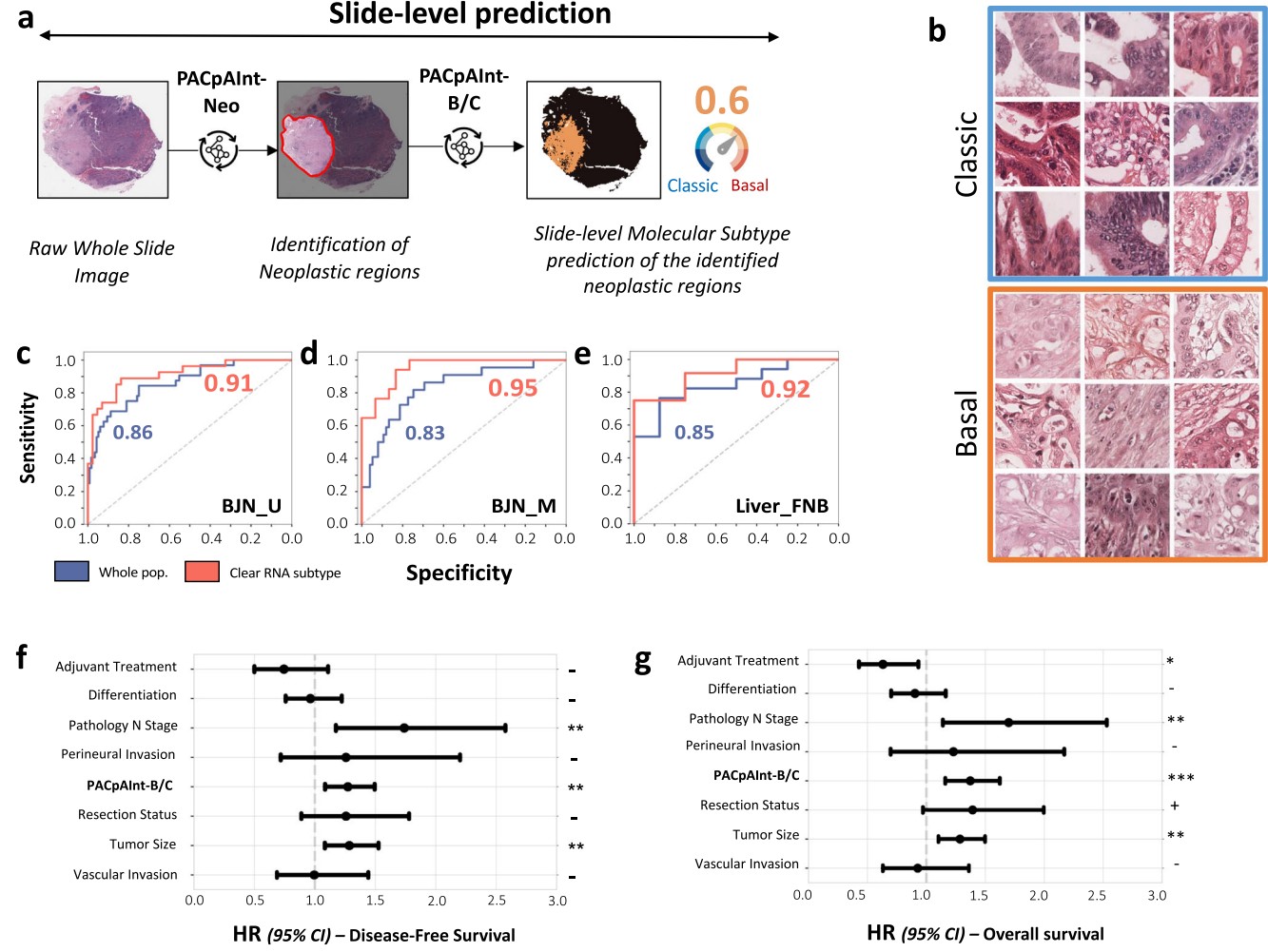

**Fig. 2 | External validation of molecular subtypes PACpAInt-B/C and association with survival. a** Simplified workflow of the study: a first model is applied to find the tumor (PACpAInt-Neo) followed by a second model predicting the global tumor cell molecular type (classical vs basal-like) at the slide level (PACpAInt-B/C), **b** Representative tiles identified as classical or basal-like by PACpAInt-B/C in the validation BJN_U cohort (112 µm square), **c**–**e** Performance of PACpAInt-B/C to identify molecular subtypes using the whole cohort or only cases with an unambiguous RNA subtype (clear subtype) of the validation cohorts (BJN_U unmatched (surgical specimens), i.e., slide analyzed and tissue used for RNAseq are not

spatially matched; BJN_M matched (surgical specimens), i.e., slide analyzed and tissue used for RNAseq are spatially matched; Liver_FNB (EUS fine-needle biopsies), **f, g** Multivariate analyses of clinical/pathological factors and PACpAInt-B/C demonstrating an independent prognostic value of the later on disease-free survival ($n = 243$) and overall survival ($n = 248$). The circle represents the variable hazard ratio, while whiskers represent the 95% confidence interval of that hazard ratio. *P* values were computed using a two-sided Wald test. No adjustments for multiple comparisons were made. ***$P < 0.001$; **$P < 0.01$; *$P < 0.05$; +$P < 0.1$; −$P > 0.1$. Source data are provided as a Source Data file.

cohorts BJN_U and TCGA (Fig. 4b, c). PACpAInt-Cell Type was further validated on a subset of the BJN_M cohort ($n = 50$) for which the tumor cells/stroma ratio was digitally computed using tumor-specific pan-cytokeratin immunohistochemistry (Pearson's $R = 0.72$, $p < 0.001$) (Fig. 4d). Using this model's predictions on a pooled cohort (DISC, BJN_M and BJN_U, $n = 451$), we confirmed that a high amount of stroma was independently associated with a better prognosis (HR = 0.86 [0.76−0.96], $p = 0.01$ and HR = 0.87 [0.77−0.98], $p = 0.02$ for DFS and OS, respectively) (Fig. 4e and Supplemental Table 4), as formerly reported[16–18].

**PDAC displayed major intratumor microheterogeneity with an important prognostic impact**
We then applied the complete three-step model on every tile (112 µm wide square) of every slide to predict whether it contains neoplastic tissue, tumor cells or stroma and their molecular phenotype (mean nb of tiles/slide = 23,306) (Fig. 5a). Concordance between tile level predictions of the model (basal-like/classical on tumor cell tiles and active/inactive on stroma tiles) and tile scoring by two expert pathologists in

PDAC was good (concordance = 100 and 99.2% for tumor components and 99.2 and 99.4% for stroma components). In addition, tiles predicted to have a basal-like and classical phenotype according to PACpAInt-Comp were tiles containing tumor cells (according to PACpAInt-Cell Type). Similarly, tiles predicted as containing active or inactive stroma (stromaActive or StromaInactive component high score, respectively) according to PACpAInt-Comp were tiles containing stroma (and no tumor cells) according to PACpAInt-Cell Type (Fig. 5c). PACpAInt-Comp was further evaluated on slides stained with GATA6/Claudin18 and KRT17 antibodies, three established specific markers for classical and basal-like phenotypes respectively[19,20]. PACpAInt-Comp was able to discriminate between GATA6 + /Claudin18+ versus KRT17+ areas, with AUCs of 0.87 and 0.75 for basal-like and classical tumor scores, respectively (Fig. 5d). Using PACpAInt-Comp predictions, we also highlighted a strong association between active stroma and basal-like rather than classical phenotypes (Fig. 5e) as previously reported[5]. In a multivariate Cox model, the use of the PACpAInt-Comp scores significantly improved the prognosis prediction with respect to clinico-molecular data alone (+4 c-index, $p = 0.007$

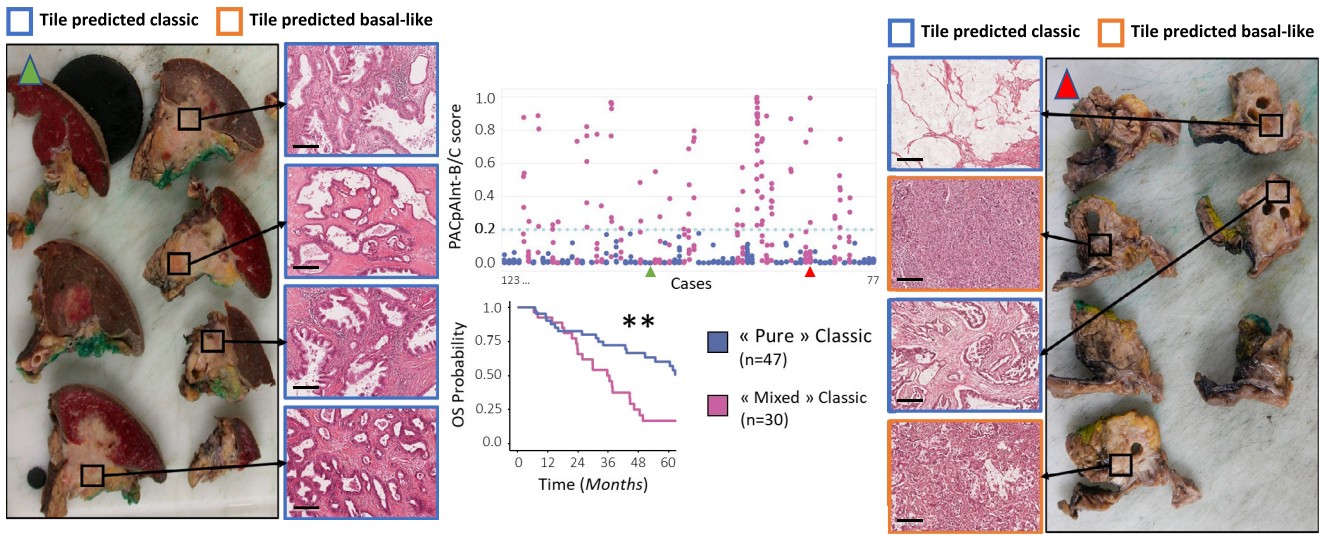

**Fig. 3 | Intra-tumoral macroheterogeneity identified by PACpAInt with whole tumor analysis.** For 77 cases defined as classical by RNAseq, all the histological slides containing tumor were digitized (n = 660) and classified by PACpAInt-B/C. Top center panel: The PACpAInt-B/C score estimating the "basalness" of each slide is represented on the Y axis while patients (1 to 77) are lined along the X axis. Each spot represents a slide. Cases with all their slides showing a low PACpAInt score (<0.2) were called "pure" classical compared to more heterogeneous tumors called "mixed" classical because at least one slide was predicted to be basal-like. Bottom center panel: Kaplan–Meyer analysis of overall survival comparing "pure" and "mixed" classical tumors (p value = 0.001538). ***p < 0.001; **p < 0.01; *p < 0.05; +p < 0.1; −p > 0.1. Left panel: Case identified to be "pure" classical by PACpAInt

(green arrowhead on the central panel). Macroscopic images of the resection showing where the tissue was sampled and the corresponding histological aspect (scale bar = 200 µm). While the areas were spatially distant, the tumor morphology was homogeneous, featuring a gland-forming pattern and good differentiation across the whole tumor. Right panel: Case identified as a "mixed" Classic by PAC-pAInt (red arrowhead on the central panel). Macroscopic images of the resection showing where the tissue was sampled and the corresponding histological aspect. Here the morphology is highly heterogeneous with spatially distinct gland-forming and non-gland-forming areas. p values were computed using a two-sided log-rank test. Source data are provided as a Source Data file.

and +3 c-index, p = 0.008 for OS and DFS respectively) (Supplemental Table 5).

## PACpAInt highlights PDAC intratumor microheterogeneity and its prognostic impact

We further assessed the impact of intratumor heterogeneity, using PACpAInt-Comp to spatially phenotype a total of 6.3 million tumor tiles encompassing 451 patients of the pooled cohorts (DISC, BJN_U, and BJN_M). PACpAInt subtyping revealed that 71% of tumors presented a detectable fraction of clearly basal-like tumor cells, confirming that most PDAC do contain basal-like cells as suggested by previous single-cell analyses on 12 cases[8,21]. The overall proportion of basal-like cells was prognostic, with worsened prognosis starting at 5% of clearly basal-like identified tumor cells and was independently associated with OS and DFS in a multivariate analysis (Fig. 6a, b and Supplemental Table 6). In addition, as the proportion of basal-like tile increased, the proportion of Inactive stroma tile decreased while that of Active stroma increased (Fig. 6c).

The tile-by-tile analysis of the basal-like and classical PACpAInt-Comp scores showed that only 60% of tumors had an unambiguous main subtype (classical 41% and basal-like 19%). The remaining could be divided into an infrequent hybrid subtype (10%) defined by the coexistence of both clearly differentiable basal-like and classical tumor cells, and an intermediary subtype (30%) for which most tumor cells could not be clearly assigned to any of the two subtypes (Fig. 7a, b). Further supporting these findings, the RNA basal-like signature was high in cases predicted by PACpAInt to be basal-like, low in classical and intermediate tumors (Fig. 7c). The opposite was found with the classical RNA signature. Interestingly, tumors predicted to be hybrid displayed high classical and basal-like signatures confirming their dual nature. The subtype prediction by PACpAInt had a strong prognostic impact. As expected, the main classical tumors had the best prognosis (median DFS 27.5 months, median OS 45.1 months) and main basal tumors the worst (median DFS 8.4 months, median OS 13.6 months)

(Fig. 7d and. Supplemental Table 7). Intermediary and hybrid tumors showed an intermediate prognosis (median DFS 15.7 months, median OS 33.0 months for intermediary tumors and median DFS 12.3 months, median OS 23.4 months for hybrid tumors). Distribution of cases based on their transcriptomic profile in principal component analysis clearly showed that AI-defined intermediary tumors are true in-between lesions (Fig. 7e). Gene set variation analysis showed that AI-defined intermediary tumors, like basal-like tumors, shut down gastric/intestinal-like differentiation programs, hallmarks of classical tumors but did not upregulate basal-like signatures (squamous and EGFR/KRAS pathways) confirming their intermediate nature (Fig. 7f).

## Discussion

With a global consensus on PDAC molecular subtypes finally emerging and early results suggesting their potential predictive value in addition to a strong prognostic value, the need for efficient and reliable tumor subtyping is greater than ever[6]. In this study, we developed PACpAInt, an AI-based tool able to predict on routine pathology slides PDAC molecular subtypes of both tumor and stromal cells. Our approach relies on an interpretable deep-learning design, translating molecular signatures defined on whole tumors into morphology-based spatialized cell phenotyping for comprehensive intratumor heterogeneity analyses.

Our training cohort included slides from different centers over a long period of time with different staining protocols ensuring a wide variability in stainings to build robust models. The validation of the models on four independent cohorts is a strength of this study. The good performance on the validation cohorts from a fourth center and the multicentric curated TCGA_PAAD cohort also supports the robustness of the models, especially since the TCGA_PAAD slides were stained with H&E while the rest of the slides were stained with H&E + Safran, the standard histological coloration in France that highlights better the fibrotic stroma in some cases. We also performed spatial validation of the tumor/stroma subtypes model using IHCs to define

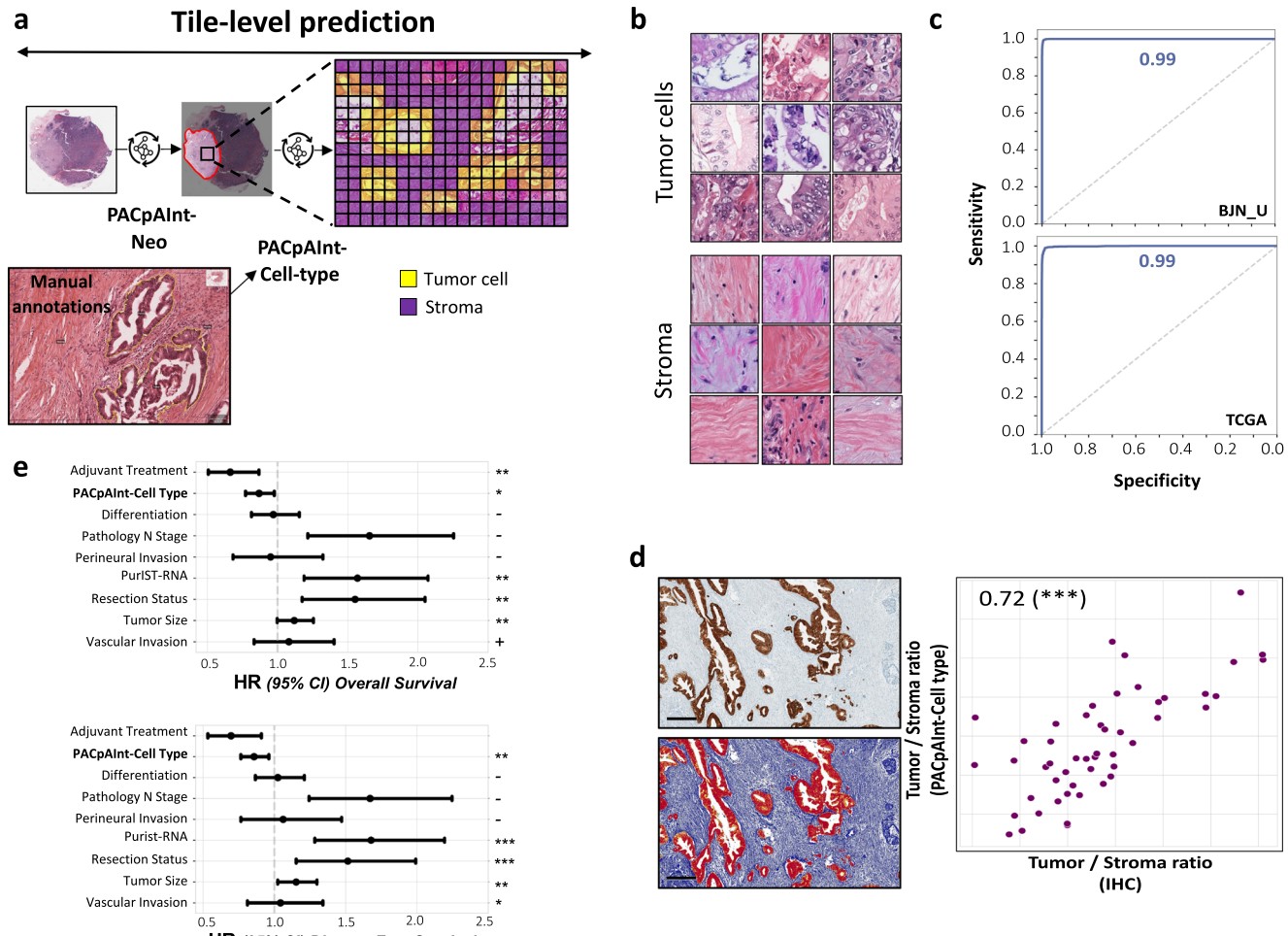

**Fig. 4 | Identification of tumor cells and stroma by PACpAInt-Cell type.**
**a** Workflow for cell type identification with PACpAInt-Cell type. PACpAInt-Neo is first applied to identify neoplastic regions, followed by PACpAInt-Cell type, which, inside these regions, distinguishes tumor cells from the stroma. PACpAInt-Cell type was trained on regions of 81 slides of the DISC cohort, annotated by two expert pathologists at the cell level. **b** Representative tiles identified as tumor or stroma by PACpAInt-Cell type in the TCGA validation cohort (112 μm square). **c** Performance of PACpAInt-Cell type to identify tumor and stroma cells in the BJN_U (top) and TCGA (bottom) validation cohorts, **d** correlation between the tumor cell/stroma ratio computed by PACpAInt-Cell type or with a computer-assisted calculation of the tumor stroma/ratio based on pan-cytokeratin immunohistochemistry (scale bar = 300 μm) (ratio stained area/total tumor area) ($p$ value = 3.16e-09). $p$ values were computed using a two-sided $t$-test, **e** Multivariate analyses of clinical/patho-logical factors and PACpAInt-cell type computed tumor/stroma ratio on disease-

free (left, $n = 428$) and overall (right, $n = 451$) survival in the pooled DISC cohort and BJN_U + M validation cohorts. The circle represents the variable hazard ratio, while the whiskers represent the 95% confidence interval of that hazard ratio. $P$ values for overall survival are: adjuvant treatment, $p = 0.001935$; PACpAInt-cell type, $p = 0.020405$, perineural invasion, $p = 0.797476$; differentiation, $p = 0.731345$; vas-cular invasion, $p = 0.512674$; Pathology N stage, $p = 0.001960$; PurIST-RNA, $p = 0.001058$; resection status, $p = 0.002100$; tumor size, $p = 0.053910$. $P$ values for disease-free survival are: adjuvant treatment, $p = 0.007799$; PACpAInt-cell type, $p = 0.008226$, perineural invasion, $p = 0.721716$; differentiation, $p = 0.779395$; vas-cular invasion, $p = 0.745392$; pathology N stage, $p = 0.000665$; PurIST-RNA, $p = 0.000153$; resection status, $p = 0.002868$; tumor size, $p = 0.019464$.***$p < 0.001$; **$p < 0.01$; *$p < 0.05$; +$p < 0.1$; −$p > 0.1$. Source data are provided as a Source Data file.

basal-like and classical regions on the H&E slides and pathologist annotations to define Active and Inactive stroma regions as gold standards. Most importantly, we validated the tumor subtype identi-fication on liver biopsies, the most common diagnostic samples for PDAC diagnosis. PACpAInt performed equally well even when decreasing of 25% the number of analyzed tiles by suggesting that it might be useful even when the amount of tumor tissue is scarce. Our study does not bring any new deep-learning methods but rather relies on existing deep-learning techniques that have proven to work well in a variety of tasks[22,23]. PACpAInt brings RNA-free PDAC molecular sub-typing and the analysis of PDAC intra and intertumor heterogeneity at a large scale with deep learning.

While the binary classification of tumor cells fails to faithfully recapitulate the complexity of PDAC, it provides a tool that is easy to implement to help appreciate the prognosis and potentially decide the

treatment instantly, without the lengthy and costly RNAseq analysis, allowing its use in clinical trials to stratify patients. RNA-based strati-fication of tumors can be dramatically impaired in samples with few tumor cells and/or heavily contaminated by non-tumor cells, a very frequent situation in pancreatic biopsies. In addition to these chal-lenges, predictive RNAseq in routine practice in PDAC would need to be performed very rapidly to avoid delaying the first administration of chemotherapy. This means that unless a large effort of centralization is undertaken, each center would process a few samples a week, leading to high cost and difficulties in sample normalization. PACpAInt used at the slide level could help overcome some of these problems. In addi-tion, it may help pathologists of small centers with no access to RNAseq. A clear limitation of PACpAInt will be the amount of tissue that can be analyzed on a biopsy. Very small lesions leading to few cells, sufficient for the diagnostic of malignancy, may not be adequate

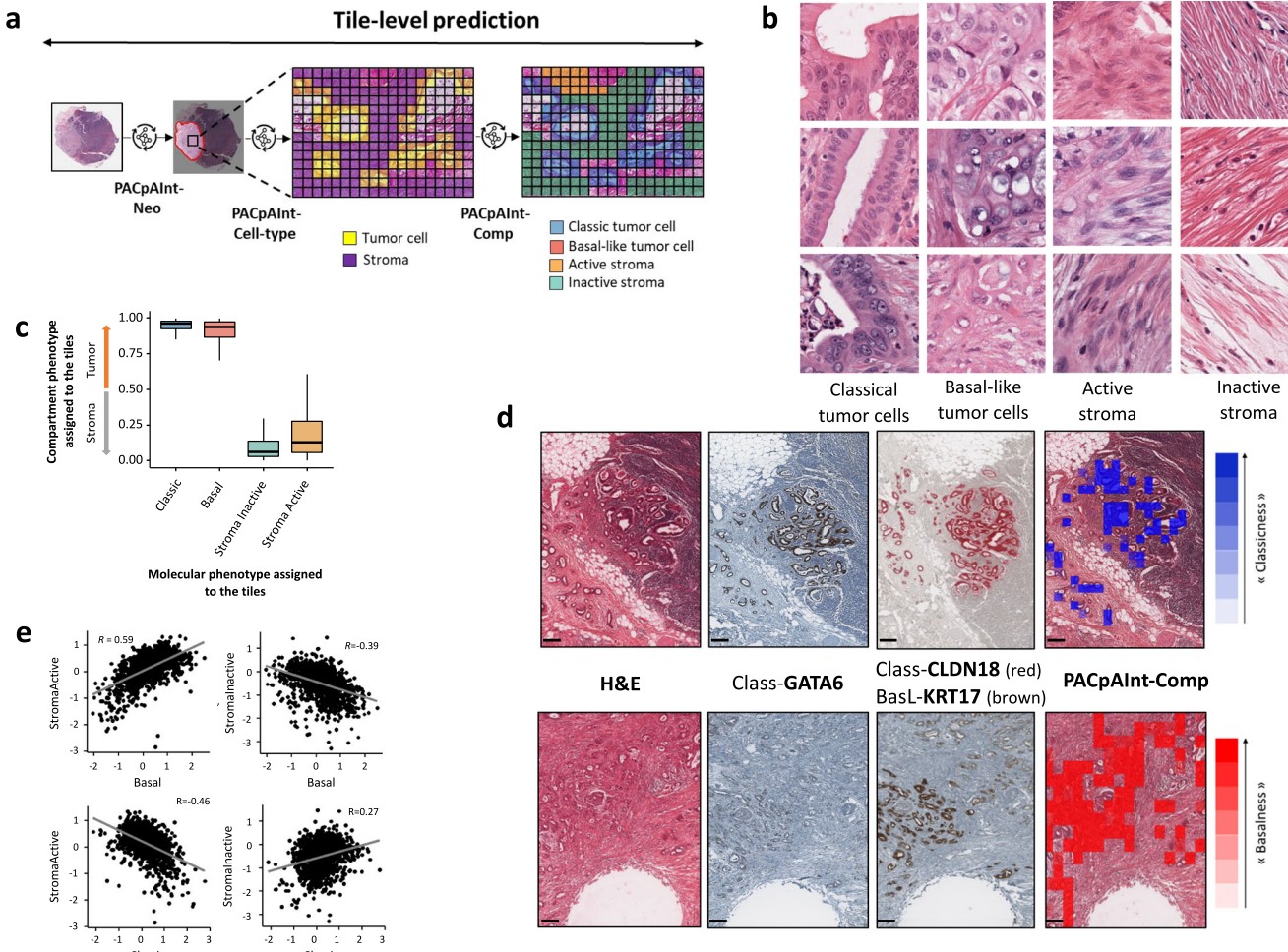

**Fig. 5 | PACpAInt identification of molecular components to decipher intra-tumor microheterogeneity. a** Workflow for molecular components identification at tile-level. PACpAInt-Neo is first applied to identify neoplastic regions, followed by PACpAInt-Cell type to identify tumor cells and stroma, then PACpAInt-Comp to predict the molecular subtype of tumor cells and stroma, **b** Representative tiles identified as tumor classical or basal-like or stroma active or inactive by PACpAInt-Comp in the TCGA validation cohort (112 μm square), **c** PACpAInt-Cell type tumor and stroma score in tiles identified as classical, basal-like, stroma active or inactive by PACpAInt-Comp (analysis on 100 K tiles for each category (i.e., classic, basal, etc.). Center corresponds to the median, lower, and upper hingers to the first and third quartiles, whiskers to the hist/lowest value no further than 1.5 × IQR (inter-quartile range), **d** Example at the tile level of areas identified as classical or basal-like by PACpAInt-Comp and stained by immunohistochemistry with classical (GATA6/Claudin18) or basal-like (KRT17) markers (scale bar = 200 μm), **e** Correlation between slide-wise median stromal and epithelial scores. Source data are provided as a Source Data file.

for PACpAInt. Deep-learning models applied to CT scans could represent an attractive non-invasive alternative. So far, models were proposed for PDAC diagnosis (vs autoimmune pancreatitis) or to predict simpler molecular labels that were reported to impact survival in PDAC, such as KRT81 positivity by immunohistochemistry[24,25]. In addition, deep-learning models like PACpAInt-Neo could be used to detect remaining tumor cells after neoadjuvant treatment, paving the way for a standardized regression score that could also be used in trials to adjust adjuvant therapy[26,27]. This is of particular interest as a recent study from an international group of experts highlighted the lack of inter-observer concordance to grade the neoadjuvant tumor response using the CAP score[28]. In a full digital pathology lab, this type of model would be quicker than cytokeratin staining with computer-assisted counting of the tumor density.

PACpAInt allowed us to assess intratumor heterogeneity at a deep scale. Few studies performed multi-areas RNAseq, each on a small number of cases, suggesting that the two main subtypes may be present in a single tumor[29,30]. Our results provide a clear picture of PDAC intratumor heterogeneity, showing that almost a third of the tumors are likely halfway between the classical and the basal-like subtypes. This is of major interest as several epigenetic drugs are being

developed to try to reprogram PDAC cells. Our data showed that a minor basal-like component, that would be ignored by binary classifications, has a strong prognostic implication. Finally, this study also demonstrates that the stromal compartment can be rapidly subtyped, paving the way for patient stratification in drug-targeting trials. Deep-learning models are often criticized for being black box models that lack explainability. Models used in our study are interpretable by design and we were able to retrieve the most predictive regions. Their analysis by pathologists confirmed the biological meaningfulness of the model findings. Furthermore, weakly supervised approaches like the ones we used for tumor / stroma subtype identification can be used to highlight local patterns (i.e., tumor areas with specific features like necrosis, presence of tertiary lymphoid structures, etc.) that could explain an otherwise unclear global molecular label, possibly uncovering key cellular components and/or their interactions.

In conclusion, while demonstrating the value of histology-based deep-learning models for tumor subtyping in PDAC, these results also show the limit of molecular-based subtyping in highly heterogeneous samples. With the expansion of digital pathology, remote AI-based PDAC subtyping could be deployed worldwide, finally opening the way for patient stratification based on powerful molecular criteria.

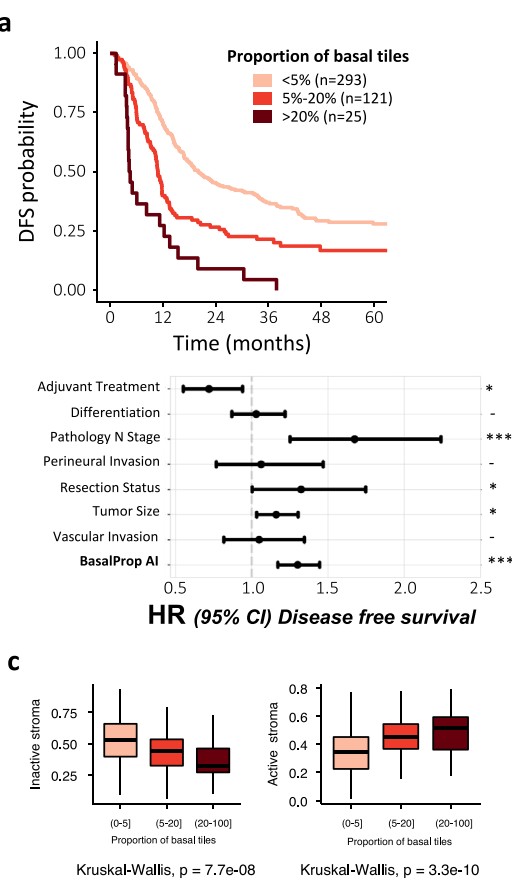

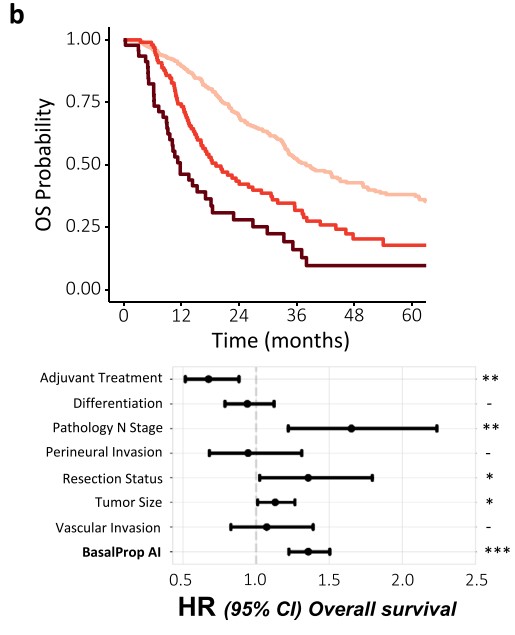

**Fig. 6 | Estimating the impact of a minor basal-like tumor component.**
**a** Kaplan–Meier and multivariate analysis of the disease-free ($n = 428$) and **b** overall survival ($n = 451$) comparing tumors with less than 5%, 5 to 20%, and more than 20% of their tumor tiles being identified as basal-like, **c** Association between the percentage of the basal-like tiles and the proportion of inactive or active stromal tiles in the same tumors ($n = 451$ in total, [0–5] $n = 305[5–20]$, $n = 121[20–100]$, $n = 25$). Center corresponds to the median, lower, and upper hingers to the first and third quartiles, whiskers to the hist/lowest value no further than $1.5 \times IQR$. Source data are provided as a Source Data file.

## Methods

### Ethical compliance
This study (ref 2020-013) was reviewed and approved by the "Comite d'Evaluation de l'Ethique des projets de Recherche Biomedicale (CEERB) Paris Nord" (Institutional Review Board -IRB 00006477- of HUPNVS, Paris 7 University, AP-HP). Non-deceased patients were informed in writing of the study. According to the French Jardé Law for non-interventional studies, they had a 2-month period to express in writing their opposition to the study and were otherwise considered as willing participants. Patients were not compensated for their participation in the study.

### Datasets description
This study (ref 2020-013) was reviewed and approved by the "Comite d'Evaluation de l'Ethique des projets de Recherche Biomedicale (CEERB) Paris Nord" (Institutional Review Board -IRB 00006477- of HUPNVS, Paris 7 University, AP-HP). The discovery set (DISC cohort) used to develop our models is a multicentric cohort of 202 consecutive patients treated in three different centers between September 1996 and December 2010: Saint-Antoine University Hospital, Pitie-Salpetriere University Hospital or Ambroise Pare University Hospital. At least two hematoxylin-eosin ± Safran (HES) slides from surgical specimens were available for each patient, corresponding to a total of 424 slides.

BJN_U and BJN_M are two independent validation cohorts of patients treated at a fourth center, Beaujon University Hospital, between September 1996 and January 2014. BJN_U consists of 304

HES slides of consecutive surgical resection specimens from 148 patients. For all the cohorts above, a punch (0.8 mm diameter core, see below in the transcriptome section) was made in a single block in an area rich in tumor cells. The slides that were digitized from these tumors may or may not come from the same block as the one punched for the RNA extraction, i.e., the "spatially unmatched" nature of these cohorts. In contrast for the "matched" cohorts, BJN_M and Liver_FNB cohorts, one block was selected and the complete tumor area of that block was microdissected for RNA extraction on serial sections with the HES. The extracted RNA is therefore coming from the same block as the HES, but most importantly, the RNA is not coming from a small portion of the block but from the whole tumor area corresponding better to the whole HES slides. To study intratumor heterogeneity, we also digitized for the BJN_M cohort all the additional slides with tumor cells on them, corresponding to a total of 909 HES slides for 97 patients. Liver_FNB is a third independent validation cohort of endoscopy ultrasound fine-needle biopsies from a liver metastasis of 25 patients (one biopsy per patient) treated at Beaujon University Hospital between 2013 and 2020. The median size of the metastases sampled was 20 mm IQR [5–15].

TCGA_PAAD is a multicentric independent validation cohort of 134 hematoxylin-eosin (H&E) slides (126 cases) from a public dataset of the TCGA database[31]. Similarly to the DISC and BJN_U cohorts, for each patient, the area sampled and frozen that was subsequently used for RNA extraction is different from the H&E slide that was analyzed (i.e., this cohort is also "spatially unmatched").

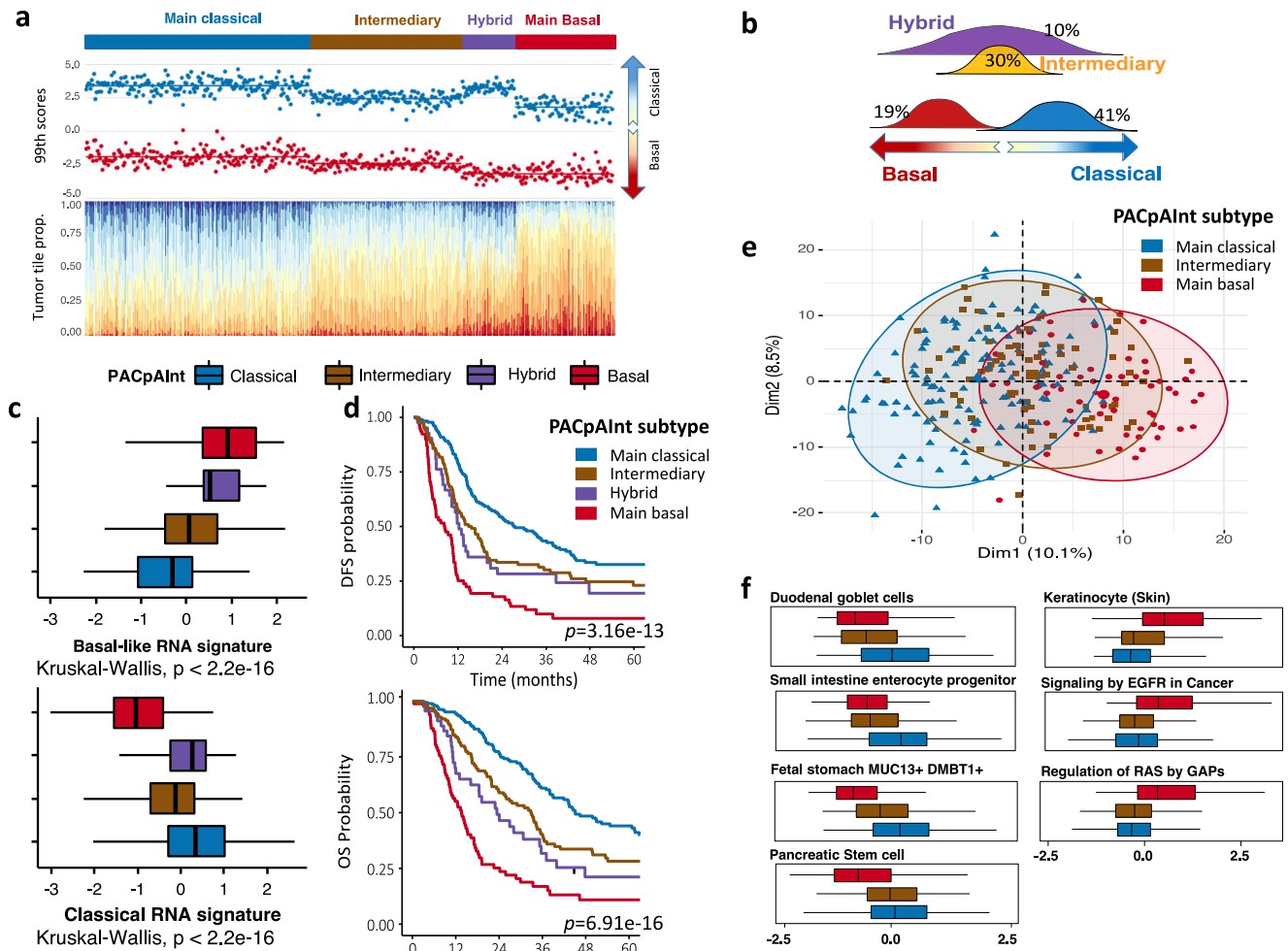

**Fig. 7 | PACpAInt identification of intra-heterogeneity-based PDAC subtypes.**
**a** Patient distribution in four subtypes: main-classical, intermediary, hybrid, and main basal-like. For each column-wise patient is first shown the 99th percentile basal-like and classical scores and second the proportion of tumor tiles for different levels of basal-like and classical phenotype, **b** Schematic illustration of the intra-tumoral distribution of tumor cell phenotypes along a basal-like vs classical differentiation axis, **c** Distribution of the tumor-level RNA-defined signature scores of the basal-like and classical phenotypes compared between the four subtypes: main classical, intermediate, hybrid, and main basal-like, (*n* = 451 in total, high.bas *n* = 84, high.cla *n* = 192, hybrid *n* = 45, inter *n* = 130). Center corresponds to the median, lower and upper hingers to the first and third quartiles, whiskers to the hist/lowest value no further than 1.5 × IQR, **d** Kaplan–Meier analysis of disease-free and overall survival comparing main classical, intermediate, hybrid, and main basal-like tumor, **e** Distribution of PACpAInt-defined subtypes in a principal component analysis based on transcriptomic profiles of the tumors, **f** Differential enrichment by gene set variation analysis of pathways in PACpAInt-defined subtypes (*n* = 272 in total, high.bas *n* = 61, high.cla *n* = 128, inter *n* = 83). Center corresponds to the median, lower and upper hingers to the first and third quartiles, and whiskers to the hist/lowest value no further than 1.5 × IQR. Source data are provided as a Source Data file.

Inclusion criteria for all cohorts were as follows: unequivocal diagnosis of the most common histological variants of pancreatic adenocarcinoma (i.e., ductal, adenosquamous, and colloid carcinomas), available histological slides of formalin-fixed, paraffin-embedded material, available follow-up, and molecular information, absence of metastasis at diagnosis. Adenosquamous and colloid carcinomas are bona fide duct-derived neoplasms that are close to the extreme of the RNA classification but are still within the range of the other PDAC when classified by the PurIST tool (Suppl Fig. 4). Very rare subtypes for which there is to date very little information regarding their molecular biology or their cell of origin were discarded (undifferentiated carcinoma with/without osteoclastic-like cells, hepatoid carcinoma). This led to the exclusion of 34 slides from the TCGA that had either no tumor cell on the slide or were from frozen examinations. The pathological information were derived from the pathological reports. The routine analysis of the included surgical specimens was performed by 4 expert pancreatic pathologists (Training cohort: M. Svrcek (center 1), A. Bardier Dupas (center 2), J.F Emile (Center 3) / Validation cohorts: J. Cros). The grading was performed according to the WHO guidelines except

for the mitosis count, that is not part of the American or French pathology guidelines for PDAC. Tumors are therefore presented as well/moderately and poorly differentiated rather than the WHO G1-G4. This study was performed according to the TRIPOD guidelines (see the TRIPOD form attached, Supplemental Table 8). The test and validation cohorts were comparable (Supplemental Table 9).

**Transcriptome profiling and molecular subtyping**
The discovery cohort corresponds to 202 resected tumors from the Puleo et al. study, which were profiled using U219 Affymetrix microarrays (GEO accession number: GSE85916). For the BJN_U cohort, RNA was extracted from a 0.8 mm diameter core sampled from a tumor-enriched zone. In most cases, the RNA was not extracted from the same block that was used to generate the HES slides. For the BJN_M and Liver_FNB series, RNA was not extracted from a small punch but after manual microdissection of two serial slides to remove contaminating normal liver or pancreatic tissue. This means that the RNA is the reflection of the whole tumor area of the block and corresponds exactly to the analyzed HES. In addition, for the BJN_M cohort, all the

other tumor slides were also analyzed by PACpAInt. For the BJN cohorts, DNA/RNA was extracted using the ALLPrep FFPE tissue kit (Qiagen, Venlo, The Netherlands) following the manufacturer's instructions and sequenced using 3' RNAseq (Lexogene Quantseq 3'). RNAseq reads were mapped using STAR v2.7.5a and genes were quantified using FeatureCount (data available here: 10.5281/zenodo.7716782). Gene counts were upper quartile-normalized and logged. PurIST-RNA was applied to both microarray and RNAseq profiles resulting in a class label for each sample. The tumor and stroma components were applied to both microarray and RNAseq profiles resulting in a continuous score for each component in each sample, as previously reported[32]. For each dataset, the difference between the scaled basal-like and classical component scores were computed, and samples that had a difference above the median were considered to have a clear RNA subtype. Pathway activation was measured by the Gene Set Variation Analysis (GSVA) with a Poisson kernel and using the C8 MSigDB and Reactome gene sets.

### Preprocessing of whole-slide images

The application of deep-learning algorithms to histological data is a challenging problem, particularly due to the high dimensionality of the data (up to $100,000 \times 100,000$ pixels for a single whole-slide image) and the small size of available datasets. Therefore, a preprocessing pipeline composed of multiple steps was used to reduce dimensionality and clean the data. The first step consists in detecting the tissue on the WSI: a U-Net neural network is used to segment part of the image that contains matter, and discard artifacts such as blur, pen marker, etc., as well as the background[33]. This U-Net network was previously trained on 460 H&E and IHC slides from an internal dataset where the tissue was manually annotated and validated on 115 slides with a Dice score of 0.96. The second step consists in tiling the slide into smaller images, called "tiles," of $112 \times 112$ μm ($224 \times 224$ pixels). At least 50% of the tile must have been detected as foreground by the U-Net model to be considered as a tile of matter. The final step consists of extracting features from each tile; color histogram normalization is applied, and 2048 relevant features are extracted using a wide Resnet50 network (the bottleneck number of channels is twice larger in every block[34], trained in a self-supervised fashion MoCo v2, using the approach proposed by Dehaene et al.[35]. This network was trained on 4 million tiles from TCGA-COAD dataset, with massive data augmentation (random cropping, random flips, color jitter, random grayscale, and gaussian blur), and without using any labels. Feature extractor weights were frozen both for inference and training. At the end of this preprocessing pipeline, each slide is represented by a matrix of size ($n_{tiles}$, 2048).

### Neoplastic and cell type prediction

PACpAInt neoplastic prediction model (PACpAInt-Neo) was trained at the tile level, based on the exhaustive neoplastic annotations of 433 slides from the discovery cohort provided by two expert pathologists, which corresponds to a total of 9,886,596 tiles. Slides were annotated using the Aperio Imagescope software v.12.4.6 (Leica™). WSI preprocessing, described in the section "Preprocessing of whole-slide images" was used to obtain 2048 features for each tile. PACpAInt-Neo architecture consists of a multi-layer perceptron with a single layer of 128 hidden neurons, followed by ReLU activation. The model was trained on non-exhaustive annotations of tumor cells and stroma using the Aperio Imagescope software v.12.4.6 (Leica™) and validated on regions annotated by two pathologists of slides of the cohort BJN_U and TCGA (ten slides for each cohort). PACpAInt-cell type prediction model (PACpAInt-Cell type) has the same architecture as PACpAInt-Neo, and was trained on annotations of 81 slides from DISC cohort, which corresponds to a total of 66,920 tiles. Likewise, it was validated on regions of ten slides of BJN_U and TCGA_PAAD annotated by two pathologists (Aperio Imagescope software v.12.4.6 (Leica™)).

### Molecular prediction

PACpAInt-B/C and PACpAInt-Comp are two multiple instance learning models that were trained on the discovery cohort at the slide level to predict respectively PurIST-RNA basal-like classification and the molecular components Classical, Basal-like, StromaActiv, StromaInactive. The two models use the same WSI preprocessing pipeline described in the section "Preprocessing whole-slide images," but PACpAInt-Neo was further applied to the tile features in order to select only tiles in neoplastic regions (i.e., with a neoplastic prediction score larger than 0.5). During training, a maximum of 8000 tiles are uniformly sampled from each slide for speed and memory considerations. For inference, all tiles are used. PACpAInt-B/C architecture is similar to the one proposed by Ilse et al. A linear layer with 128 neurons is applied to the tile features followed by a Gated Attention layer with 128 hidden neurons[22]. We then apply a multi-layer perceptron (MLP) with 128 and 64 hidden neurons and ReLU activations to the results. A final Sigmoid activation is applied to the output to obtain a score between 0 and 1, which represents the probability of the slide to be basal-like or classical. PACpAInt-B/C was trained with the binary cross entropy as loss function, using PurIST-RNA basal-like classification defined by RNA sequencing at patient-level as labels. PACpAInt-Comp was inspired by the WELDON algorithm: 4 scores for each tile are computed from the tile features, where each score corresponds to each of the 4 molecular components[23]. This scoring is performed using a MLP with 128 hidden neurons followed by four neurons and ReLU activation. For each score dimension, we select $R = 100$ top and bottom scores and average them, so that the model's output is a vector of size 4, corresponding to the continuous predicted values of each molecular component. The model was trained with the mean squared error as a loss function, using molecular components defined by RNA sequencing at patient-level as labels.

### Spatial validation

To validate locally the accuracy of PACpAInt-Comp to predict classical and basal-like, GATA6, Claudin18, and KRT17 IHCs were performed on 12 slides of BJN_M. The following antibodies were used (GATA6 (Cell Signaling, clone D61E4, Rabbit, at 1/200, REF5158S), Claudin18 (Sigma, Polyclonal ref HPA018446, Rabbit, at 1/50, REF HPA01846), KRT17 (BioSB, clone BSB-33, mouse, at 1/800, REF BSB2729), PanCK+ (Zytomed, clone coktail AE1/AE3/5D3, mouse, at 1/300, REF MSK098-05). Immunohistochemistries were performed on a Ventana (Tuscon AZ, USA) benchmark ultra automates. Tile scores for classical and basal-like components were analyzed in regions defined as being basal-like/classical by the IHCs. Two expert pathologists also analyzed tiles predicted to be classical or basal-like ($n = 500$) and tiles predicted to be stroma active or inactive ($n = 500$), blinded to scores associated with each tile.

### Performance assessment and statistical methods

The area under the receiver operating characteristic curve (AUC) was used to quantify the capability of the model to distinguish classical from basal-like tumors, as assessed by the PurIST method. The same metric was used to assess the performance of PACpAInt-Neo to distinguish normal from neoplastic regions, and of PACpAInt-Cell-Type to distinguish stroma from epithelial tumor cells. Delong's method was used to compute confidence intervals at 95% confidence level[36]. Pearson's correlation was used to assess the performance of the PACpAInt-Comp model to predict the molecular components. Survival analyses were performed with uni- and multivariate Cox proportional hazards models implemented in the lifelines package of Python[37]. Log-rank tests were used to compare survival distributions between population subgroups. We used survcomp R package to compare $c$-indexes[38]. All tests were two-tailed, and $P$ values <0.05 were considered statistically significant.

## Intratumor heterogeneity subtypes

The 99th percentile of the basal-like and classical component scores defined by PACpAInt-Comp for each patient, using all available slides per patient, were computed. The absolute difference between 99th of the basal-like and classical tiles was then used as a measure of differentiation in each patient, a high absolute difference indicating a well-differentiated (either clearly basal-like or classical) tumor. A Gaussian mixture model was applied, which identified two subgroups, one group of clearly differentiated tumors (with a higher difference between the 99th percentile of the basal-like and classical tiles) and one of the unclear tumors in which the levels of the most differentiated tiles were similar, thereby showing no clear pattern toward basal-like or classical. This latter "unspecified" subtype ($n = 175$, 38.8%) could be either composed of a mixture of well-differentiated tumor contingent or of an overall tumor phenotype neither basal-like nor classical and possibly in an intermediary differentiated state. To differentiate between these two possibilities, the maximum level of differentiation of each tumor (measured as the maximum of either the basal-like and classical 99th percentile tile level) was identified as an optimal cut-off to define clearly differentiated tumors. The idea here is to find the set of tumors that have a differentiation level as high as clearly differentiated tumors while having a low difference between basal-like and classical levels. This resulted in two partitioning systems. First, by separating by the difference in the highest level of differentiation observed, a higher difference indicating a clear subtype, the highest phenotype level indicating a main basal-like ($n = 84$, 18.6%) or a main classical tumor ($n = 192$, 42.6%). Second, by separating, among unspecific or low differences between subtypes, tumors that have both clear and highly defined basal-like and classical tumor tiles, referred to as hybrid ($n = 45$, 9.9%), and tumors that have no clearly basal-like nor classical tiles and therefore termed intermediary ($n = 130$, 28.8%).

## Clinical variables used in the multivariate analysis

Clinical variables considered for multivariate analysis were common variables known to be associated with PAC prognosis: pN stage, differentiation, perineural invasion, resection status, tumor size, vascular invasion, and adjuvant treatment yes/no.

## Reporting summary

Further information on research design is available in the Nature Portfolio Reporting Summary linked to this article.

## Data availability

The TCGA_PAAD are available from the TCGA repository (TCGA_PAAD, [https://portal.gdc.cancer.gov]). The microarray and RNAseq data can be downloaded from the GEO repository (accession number: GSE85916) and the array express repository (E-MTAB-13007) respectively. Source data are provided with this paper.

## Code availability

The packaged models can be found here: https://github.com/CharlieCheckpt/pacpaint (https://zenodo.org/badge/latestdoi/625162034)[39].

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

## Acknowledgements

This work was granted access to the HPC resources of IDRIS under the allocation AD011012519 made by GENCI. We also like to thank Ms. Colnot and Ms. Couroble for their technical help.

## Author contributions

Study conception and design: J.C., R.N., and C.S.; data collection: J.C., R.N., F.D., V.R., L.d.M., M.S., A.B.-D., J.F.E., P.H., C.N., J.B.B., N.D., J.I., and M.A.; Software: C.S., B.S., O.M., M.Z., P.C., and A.K.; analysis and interpretation of results: J.C., R.N., C.S., Y.B., M.R., Y.K., and V.P.; draft manuscript preparation: J.C., R.N., and C.S. All authors reviewed the results and approved the final version of the manuscript.

## Competing interests

Persons affiliated with Owkin own stocks in the company (C.S., B.S., O.M., M.Z., P.C., and A.K.). The remaining authors declare no competing interests.

## Additional information

¹Owkin France, Medical Imaging Team, Paris, France. ²Université Paris Cité, Dpt of Pathology - FHU MOSAIC, Beaujon Hospital, INSERM U1149 Clichy, France. ³Dpt of Pathology, Saint-Antoine Hospital - Sorbonne Universités, Paris, France. ⁴Dpt of Pathology, Pitié-Salpêtrière Hospital - Sorbonne Universités, Paris, France. ⁵Dpt of Pathology, Ambroise Paré Hospital – Université Saint Quentin en Yvelines, Paris, France. ⁶Integragen, Genomic Services & Precision Medicine, Paris, France. ⁷Université Paris Cité, Dpt of Pancreatology - FHU MOSAIC, Beaujon Hospital, INSERM U1149 Clichy, France. ⁸Dpt of Medical oncology, Paul Brousse Hospital, Villejuif, France. ⁹Medical oncology, Institut Curie, Paris, France. ¹⁰Dpt of Gastroenterology, Pitié-Salpêtrière Hospital - Sorbonne Universités, Paris, France. ¹¹Centre de Recherche en Cancérologie de Marseille (CRCM), INSERM U1068, Institut Paoli-Calmettes, Aix Marseille Université, CNRS UMR 7258 Marseille, France. ¹²Institut Génétique et Développement de Rennes (IGDR), CNRS, Université de Rennes 1, UMR 6290 Rennes, France. ¹³Techniques de l'Ingénierie Médicale et de la Complexité - Informatique, Mathématiques et Applications Grenoble (TIMC-IMAG), CNRS, Université Grenoble-Alpes, UMR5525 Grenoble, France. ¹⁴Université Paris Cité, FHU MOSAIC, Centre de Recherche sur l'Inflammation (CRI), INSERM, U1149, CNRS, ERL 8252, F-75018 Paris, France. ✉e-mail: jerome.cros@aphp.fr

