## [Peer review file · Nature Communications]

REVIEWER COMMENTS

Reviewer #2 (Remarks to the Author): Expert in pancreatic adenocarcinoma pathology and subtypes

In this work, the authors present PACpAIInt, an artificial intelligence tool, which is able to perform subtyping of pancreatic cancer according to the basal/classical and stroma inactive/active classification without the need of performing RNAseq and thus possibly overcoming the problems related to costs, feasibility, technical issues due to fixation/use of FFPE-tissues and scarcity of material, among others.

The study is relevant since molecular subtyping in PDAC is suggested to be the key for successful personalized treatment in this disease. In addition, the study has the merit to be led by pathologists, which is a fundamental requirement for this kind of studies in order to avoid misinterpretation.

There are some points that should be addressed more carefully in order to convince about the robustness of the proposed tool:

- 1) The authors should generate among their numerous collectives used as training and validation cohorts a homogenous one: only cases of PDAC NOS should be included, all homogenous variants (according to WHO criteria, e.g. mucinous, undifferentiated with and without OCG, adenosquamous) and non-ductal cancers (if any) should be excluded due to different molecular and clinical characteristics. E.g. in figure 3, right panel, there seem to be a partially mucinous PDAC included.
- 2) The authors should perform an accurate grading according to WHO 2000 criteria, including mitosis count. All classical PDAC shown in the illustrations have a clear glandular conformation (G1/G2 tumors), all basal are solid (G3). The reviewer would like to be convinced that the different characteristics and the prognostic relevance are not simply an effect of degree of differentiation. The reviewer acknowledges that the degree of differentiation, probably derived for the pathology reports, is included in the multivariate analysis. However, there is no information about how grading was done and whether it was reviewed and confirmed by the authors.
- 3) Since the authors advocate the applicability of this method to sample with low cellularity, size of the liver metastases (not of the biopsies!) should be included.
- 4) Methods: is there a role/advantage in using HES instead of H&E?

In the so-called matched cohorts was the core used for RNAseq taken before or after scanning? If this was after, then this is also not a real match (PDAC can be highly heterogeneous!) unless demonstrated otherwise (please provide representative HE prior to and after taking the core)

5) Discussion: It is not clear why the tool should be helpful to detect residual tumor after NAT. H&E stained slides and possible a pan-CK IHC are more than sufficient, easier and definitely less expensive.

Suggesting AI as substitute of experience is dangerous; the sentence “ In addition, it may help pathologist of small centers with less experience in pancreatic pathology and no access to RNAseq” should be rephrased (for example just mentioning the RNA seq) or eliminated.

6) If a pathology lab would like to implement this tool, what equipment would be necessary? What about storage space? Would a training be necessary? How many slides from a given tumor should be scanned? Are there any thoughts about implementation and practicability?

Minor comments:

The cohort BJN_M is not described in sufficient detail.

References are in some cases incomplete and/or with different styles

“Perinervous” infiltration should be changed into “perineural”

Figure legend 1: 600m is probably 600µm?

Reviewer #3 (Remarks to the Author): Expert in digital pathology in cancer and deep learning

This study is very interesting and addresses a clinically relevant problem. The study also closes a gap in the literature, as there are very few studies using artificial intelligence on histopathological images of pancreatic carcinoma.

However, there are some major and some minor issues which need to be fixed (in no particular order here)

A big point is the presentation of the results and the wording in the article. The abstract is unstructured and ends abruptly. Also, the grammatical tenses in the abstract are not consistent. I would highly

recommend dividing the abstract into logical sections, namely problem statement, methods, results, and interpretation. Also, the rest of the article basically needs to be completely rewritten to make it more clearly structured and understandable. The font size / type varies across the article. This suggests a lot of copy-and-pasting has been done and this should be fixed.

The greatest strength of the article is that four independent patient cohorts were used. And could be emphasized even more, and also the number of patients per cohort be mentioned in the abstract

Please use PDAC as an abbreviation for pancreatic carcinoma, which is much more common than PAC.

When describing the cohort it would be very very important to note that the training set and the test set was acquired from different institutions. Is this the case? If so, please state it explicitly. If this is not the case, you have a problem because the training set is contaminated with samples from the test set centers.

In the background section of the introduction, the authors should cite some work on the quality criteria for deep learning studies, such as Kleppe et al Nat Rev Cancer 2021 or Shmatko et al Nat Cancer 2022.

How did the authors perform the regression learning of continuous scores? This is not described appropriately.

The source codes should be made publicly available.

The methods section should be moved before the results section.

The results section contains a lot of non-results (methods description and background), please move this content to the appropriate sections

The authors restrict the analysis to "to the 50% of cases that had the clearest, unambiguous". This is definitely not good practice. If you want to train a classifier for cats vs. dogs, you cannot exclude the 50% most difficult cases from your dataset, and only include the "easiert" half. Of course, this procedure boosts the AUROC but I don't consider it scientifically appropriate. Please report the results for the whole cohort. If you want to use the cleaned cohort of just the easy samples, move this to the supplements.

The authors need to declare adherence to STARD or TRIPOD or an equivalent guideline by the equator network.

I am not sure that the title is backed up by the presented results.

Figure 1 has a lot of wasted white space in the top right corner, not nice.

In figure 6 a and b the rainbow colors do not convey any information and should be removed (just make the forest plot black)

Dear Editor,

We would like to thank you and the reviewers for their insightful comments that helped us improve our manuscript. Please find attached a point-by-point response to their comments and the modified version of the manuscript with the tracked modifications.

Reviewer #2

In this work, the authors present PACpAInt, an artificial intelligence tool, which is able to perform subtyping of pancreatic cancer according to the basal/classical and stroma inactive/active classification without the need of performing RNAseq and thus possibly overcoming the problems related to costs, feasibility, technical issues due to fixation/use of FFPE-tissues and scarcity of material, among others.

The study is relevant since molecular subtyping in PDAC is suggested to be the key for successful personalized treatment in this disease. In addition, the study has the merit to be led by pathologists, which is a fundamental requirement for this kind of studies in order to avoid misinterpretation.

There are some points that should be addressed more carefully in order to convince about the robustness of the proposed tool:

1) The authors should generate among their numerous collectives used as training and validation cohorts a homogenous one: only cases of PDAC NOS should be included, all homogenous variants (according to WHO criteria, e.g. mucinous, undifferentiated with and without OCG, adenosquamous) and non-ductal cancers (if any) should be excluded due to different molecular and clinical characteristics. E.g. in figure 3, right panel, there seems to be a partially mucinous PDAC included.

We agree with the reviewer that pancreatic carcinomas are highly heterogeneous with multiple variants described by the WHO. For some of these variants, there is no data on whether they are indeed of ductal origin like the undifferentiated carcinomas with and without OCG or the hepatoid variant. Undifferentiated carcinomas with and without OCG and the hepatoid variant were not part of the inclusion criteria and were not included in the series. We have added this to the M&M section (P17).

It now reads: "Inclusion criteria for all cohorts were as follows: unequivocal diagnosis of the most common histological variants of pancreatic adenocarcinoma (i.e ductal, adenosquamous and colloid carcinomas), available histological slides of formalin-fixed, paraffin-embedded material, available follow-up and molecular information, absence of metastasis at diagnosis. Very rare subtypes for which there is to date very little information regarding their molecular biology, or their cell of origin were discarded (undifferentiated carcinoma with/without osteoclastic-like cells, hepatoid carcinoma)."

For the colloid and adenosquamous carcinomas, they are bona fide duct-derived neoplasms that are close to the extreme of the RNA classification (Basal-like A and Classical B from Chan-Seng-Yue M. et al. *Nat Gen* 2020 and previous classification from Collisson *et al. Nat Gen* 2011 (Quasi-mesenchymal), Bailey *et al. Nature* 2016 (Squamous)) but are still within the range of the other PDAC when classified by the PurIST tool. To illustrate this, below is the ranking according to the PurIST score of the 100 cases of the BJN_M with matched H&E

and RNAseq with colloid and adenosquamous carcinomas highlighted demonstrating that these two morphological subtypes are well in line with the rest of the cases.

We hope this figure will convince the reviewer. Unless the reviewer believes it is very important to include this figure in the manuscript, we would prefer to leave it out for clarity. We made it clearer in the M&M section that these two histologies were considered as close variants of ductal adenocarcinoma and were included (M&M section P17, cf above).

Genomic data are also in line with these lesions being comparable with other PDAC at the exception of the activating *GNAS* mutation that is more frequent in colloid carcinomas. PACpAInt performs very well on them and we feel that while rare, they are far from exceptional and removing them would weaken PACpAInt in its ability to recognize such tumors. Of note the adenosquamous tumors are the perfect example of intra-tumor heterogeneity.

2) The authors should perform an accurate grading according to WHO 2000 criteria, including mitosis count. All classical PDAC shown in the illustrations have a clear glandular conformation (G1/G2 tumors), all basal are solid (G3). The reviewer would like to be convinced that the different characteristics and the prognostic relevance are not simply an effect of degree of differentiation. The reviewer acknowledges that the degree of differentiation, probably derived for the pathology reports, is included in the multivariate analysis. However, there is no information about how grading was done and whether it was reviewed and confirmed by the authors.

We agree with the reviewer that the tiles shown might mislead the reader into thinking that PACpAInt only recapitulates the differentiation. We therefore picked another set of predicted tiles more heterogeneous, demonstrating that the classical and the basal-like tumors may harbor multiple phenotypes and architectures (See modified Fig 2b and Suppl Fig 2b). As the reviewer pointed out, the PACpAInt classification is also an independent prognostic factor from the differentiation in the multivariate analysis.

To further demonstrate that differentiation and molecular subtypes are not completely redundant, we present in the table below the distribution of RNA-defined basal-like and classical tumors according to the differentiation. In the three validation cohorts, there were basal-like tumors in the well differentiated group and classical tumors in the poorly differentiated group while the moderately differentiated group harboring both subtypes.

We also looked at the performance of PACpAInt B/C inside the subgroups of patients with the same tumor differentiation (i.e., well, moderately and poorly differentiated) in the three validation cohorts. If PACpAInt B/C only recapitulated the histological differentiation, it wouldn't be able to discriminate between basal-like and classical tumors inside the subgroup of patients with the same differentiation, and AUCs would be close to 0.5 (random guess). As you can see in the table below, this is not the case as AUCs ranged from 0.71 to 0.90.

Cohort	Population	n	n Basal	n Classical	PACpAInt AUROC (p-value)
BJN_U	Whole population	148	32	116	0.861 [0.787-0.935] (p<10-5)
	Well differentiated tumors	75	7	68	0.834 [0.706-0.962] (p<10-5)
	Moderately differentiated tumors	47	12	35	0.771 [0.605-0.938] (p=0.00139)
	Poorly differentiated tumors	22	13	9	0.863 [0.708-1.000] (p<10-5)
BJN_M	Whole population	97	22	75	0.833 [0.734-0.932] (p<10-5)
	Well differentiated tumors	38	3	34	0.706 [0.401-1.000] (p=0.18564)
	Moderately differentiated tumors	45	12	33	0.854 [0.721-0.986] (p<10-5)
	Poorly differentiated tumors	14	6	8	0.896 [0.683-1.000] (p=0.00027)
TCGA	Whole population	126	28	98	0.806 [0.713-0.900] (p<10-5)
	Well differentiated tumors	4	1	3	NA
	Moderately differentiated tumors	61	3	58	0.776 [0.597-0.955] (p=0.00249)
	Poorly differentiated tumors	60	24	36	0.797 [0.681-0.914] (p<10-5)

Taken together, we believe this demonstrates that the molecular subtypes and PACpAInt are not just a surrogate of the histological differentiation. We added these results as supplementary table 2. It now reads in the result section P7: “To ensure that PACpAInt-B/C was not only recapitulating the histological differentiation, we assessed the performance of the model in well/intermediate and poorly differentiated tumors separately in the three surgical validation cohorts (Suppl Table 2). AUC ranged from 0.71 to 0.90, far from the random guess, confirming that differentiation and molecular subtypes are not complete surrogates of one another.”

Regarding the grading, it was indeed derived from the pathology reports. It should be noted that the grading was done by 4 expert pancreatic pathologists, all co-authors of this manuscript (Training cohort : M. Svrcek (center 1), A. Bardier Dupas (center 2), J.F Emile (Center 3) / Validation cohort: J. Cros) over the inclusion period of the study. The grading was performed according to the WHO guidelines except for the mitosis count. While indeed included in the WHO guidelines, the mitotic count is not part of the American (College of American Pathologists) or the French (Société Française de Pathologie) guidelines for pancreatic adenocarcinoma reporting and was therefore not performed. While central to the grading of neuroendocrine tumors, the mitosis count was indeed reported to have a prognostic value in PDAC (REF 1985) but appears to add little additional information to the differentiation and TNM staging (REF CAP). For transparency, we added in the M&M section that the mitosis count was not performed and that only the morphology (percentage of gland forming, cellular morphology) was used to assess the differentiation (i.e we will not use the G1_G4 scoring in the text or the figure to avoid misleading the reader). This was clarified in the M&M section (P17). It now reads “The grading was performed according to the WHO guidelines except for the mitosis count that is not part of the American or the French pathology guidelines for PDAC. Tumors are therefore presented as well/moderately and poorly differentiated rather than the WHO G1-G4.”

3) Since the authors advocate the applicability of this method to sample with low cellularity, size of the liver metastases (not of the biopsies!) should be included.

We agree with the reviewer that for very small lesions, the number of tumor cells on the biopsy may be low, although many other factors such as the experience of the radiologist/endoscopist, distance of the lesion from the skin/duodenum will also impact the number of tumor cells in the biopsies. We therefore believe that what will impact the diagnosis, regardless of its nature (histology, molecular, AI) is the number of tumor cells that can be analyzed and then the size of the lesion. We retrieved the size of the biopsied hepatic metastases from the patient records. The median size was 20mm IQR[5-15]. We included this information in the patient section and the discussion (P17). It now reads “The median size of the metastases sampled was 20mm IQR[5-15].”

As a sensibility analysis, we looked at the performance of PACpAInt B/C model on the hepatic biopsies if we subsampled the number of analyzed tiles to mimic less rich biopsies. As demonstrated below the performance of the model was similar when only 75% of the tiles were analyzed and then decreased but remained good even when only a quarter of the tumor tiles

were randomly selected for analysis confirming that the model performs well even on small biopsies.

% tumor tile used	AUC
25	0.82 [0.61-0.96]
50	0.82 [0.62-0.96]
75	0.85 [0.68-0.96]
100	0.85 [0.70-0.96]

We included this information in the result section (P8). It now reads: “To assess the sensibility of PACpAInt-B/C, we performed on the Liver_FNB cohort, a subsampling of the tiles to mimic tumor-poor biopsies. Using 75% of the tumor tiles, the AUC was similar (AUC=0.85 [0.68-0.96]) and decreased but remained good when using only 50 or 25% of the tumor tiles to predict the molecular subtypes (AUC=0.82 [0.62-0.96] and 0.82 [0.61-0.96] respectively).”

4) Methods: is there a role/advantage in using HES instead of H&E?

There is probably no advantage in using HES instead of H&E. HES is the standard stain in France in routine practice. While the safran does highlight a bit more the stroma for the pathologist, the fact that the performance of PACpAInt in the TCGA cohort (H&E stain) is similar to the French cohorts suggests that the type of stain has no impact. This may also be because the model was trained with slides from three centers that were pulled out of the archives (not restrained) and displayed a various range of “stain fading”. We added a sentence in the discussion on this matter (P13). It now reads: “The good performance on the validation cohorts from a fourth center and the multicentric curated TCGA_PAAD cohort also supports the robustness of the models, especially since the TCGA-PAAD slides were stained with H&E while the rest of the slides were stained with H&E + Safran, the standard histological coloration in France that highlights better the fibrotic stroma in some cases.”

In the so-called matched cohorts was the core used for RNAseq taken before or after scanning? If this was after, then this is also not a real match (PDAC can be highly heterogeneous!) unless demonstrated otherwise (please provide representative HE prior to and after taking the core)

There was no core taken in the matched cohorts. For these cohorts, to have an RNAseq as representative of the whole slide as possible, we macrodissected by “scratching” the whole tumor area on two serial slides with the analyzed HES to then perform RNA extraction and sequencing. The transcriptome is therefore “matched” with the image. We explained this better in the M&M (P16). It now reads: “BJN_U, BJN_M are two independent validation cohorts of patients treated at a 4th center, Beaujon University Hospital, between September 1996 and January 2014. BJN_U consists of 304 HES slides of consecutive surgical resection specimens from 148 patients. For all the cohorts above, a punch (0.8mm diameter core, see below in the transcriptome section) was made in a single block in an area rich in tumor cells. The slides that

were digitized from these tumors may or may not come from the same block as the one punched for the RNA extraction, i.e., the “spatially unmatched” nature of these cohorts. In contrast for the “matched” cohorts, BJN_M and Liver_FNB cohorts, one block was selected, and the complete tumor area of that block was microdissected for RNA extraction on serial sections with the HES. The extracted RNA is therefore coming from the same block as the HES but most importantly the RNA is not coming from a small portion of the block but from the whole tumor area corresponding better to the whole HES slides.”

5) Discussion: It is not clear why the tool should be helpful to detect residual tumor after NAT. H&E stained slides and possible a pan-CK IHC are more than sufficient, easier and definitely less expensive.

Evaluation of the residual tumor load after NAT is poorly reproducible between pathologists, especially because it is difficult to assess where the “ancient tumor bed” was and how many tumor cells remained. A recent study from an international group of experts highlighted the lack of concordance and showed that a dichotomization of the CAP does improve a bit the reproducibility ¹. PanCK IHC is a possibility, but for standardization, a computer-based quantification would be necessary requiring a pathologist-based annotation and the use of a pixel/object classifier to count tumor cells. This would in addition limit the analysis to one slide (the response rate can be heterogeneous spatially) and require a significant amount of time per case. An AI-based approach to identify tumor cells after NAT like the one described by Janssen *et al.* would provide a quantitative, fast, whole tumor cell count that could be verified by the pathologist ². As multicenter trials are being designed in which the chemo regimen after NAT is pursued or changed depending on the pathological tumor response, the latter must be very standardized and robust. This was better explained in the discussion (P14_15). It now reads: “In addition, deep learning models like PACpAInt-Neo could be used to detect remaining tumor cells after neoadjuvant treatment, paving the way for a standardized regression score that could also be used in trials to adjust adjuvant therapy ^{26,27}. This is of particular interest as a recent study from an international group of experts highlighted the lack of inter-observer concordance to grade the neoadjuvant tumor response using the CAP score ²⁸. In a full digital pathology lab, this type of model would be quicker than a cytokeratin staining with computer-assisted counting of the tumor density.”

Suggesting AI as substitute of experience is dangerous; the sentence “ In addition, it may help pathologist of small centers with less experience in pancreatic pathology and no access to RNAseq” should be rephrased (for example just mentioning the RNA seq) or eliminated.

We agree and changed the sentence keeping only the RNAseq (P14).

6) If a pathology lab would like to implement this tool, what equipment would be necessary? Slide scanner, fairly recent computer w/ wo/ GPU. What about storage space? Would a training be necessary? How many slides from a given tumor should be scanned?

The ease of implementation of this type of AI-based tool in a pathology lab would depend on at what stage of transition toward digital pathology the lab is. There is a strong trend towards the institution-driven transformation of pathology labs into full digital pathology labs

with the proper infrastructures (massive slide scanners (>400 slides), huge storage (>1000Tb) and computing network capacity, high-security measures, etc....). In this context, the integration of an AI tool is fairly easy. For a non-digital lab, in which only the slides that will be analyzed are scanned (i.e., slide for Ki-67 counting for instance), the minimum equipment is a mid-level scanner (10-50 slides capacity) capable of 20x magnification and a small server (8Tb). Regardless of the size of the lab, the routine digitized slides are usually erased every month or so, unless they are tagged by the pathologist to reduce the storage footprint.

We envisioned the use of PACpAInt as easily as dropping the digitized slides in a folder that would automatically launch the model and give the result. In our institution, PACpAInt runs on a personal mid-level computer and analyzes a slide in 2 to 5 minutes. For the simultaneous analysis of hundreds of slides in a reasonable time frame, then a dedicated GPU-equipped computer is needed. As for the training, the execution of PACpAInt can be fully automated requiring no coding experience. Results can be sent back into QUPATH for instance to be visualized.

Regarding the number of slides to be analyzed, that would be up to the pathologist to quickly glance at the morphology to decide how many slides are needed to accurately resume the heterogeneity. Yet because of the heterogeneity and the impact of a small basal-like component, we would encourage the pathologist to scan at least 3 slides/ per case, especially since it is hassle-free for him.

Minor comments:

The cohort BJN_M is not described in sufficient detail.

As mentioned above, we have expanded in the M&M the description of the matched cohort and how it was constructed (P16, cf remark above).

References are in some cases incomplete and/or with different styles.

We have corrected this but some references used for AI models relate to congress presentations and have a peculiar form.

“Perinervous” infiltration should be changed into “perineural”

We have corrected this in the text and the figures.

Figure legend 1: 600m is probably 600µm?

We have corrected this.

Reviewer #3 (Remarks to the Author): Expert in digital pathology in cancer and deep learning

This study is very interesting and addresses a clinically relevant problem. The study also closes a gap in the literature, as there are very few studies using artificial intelligence on histopathological images of pancreatic carcinoma. However, there are some major and some minor issues which need to be fixed (in no particular order here)

1) A big point is the presentation of the results and the wording in the article. The abstract is unstructured and ends abruptly.

2) Also, the grammatical tenses in the abstract are not consistent. I would highly recommend dividing the abstract into logical sections, namely problem statement, methods, results, and interpretation.

We thank the reviewer for these comments and reorganize the abstract to make it clearer while trying to match Nature communications guidelines (i.e., unstructured (no headings) and 150 words).

Also, the rest of the article basically needs to be completely rewritten to make it more clearly structured and understandable. The font size / type varies across the article.

This suggests a lot of copy-and-pasting has been done and this should be fixed.

We apologize for the typos and the problems with the front/size etc... The whole manuscript has been thoroughly reviewed and corrected by an English native speaker.

3) The greatest strength of the article is that four independent patient cohorts were used. And could be emphasized even more, and also the number of patients per cohort be mentioned in the abstract.

We added this information in the abstract (pending its length) and in the discussion (P13). It now reads: “The validation of the models on 4 independent cohorts is a strength of this study.”

4) Please use PDAC as an abbreviation for pancreatic carcinoma, which is much more common than PAC.

We agree with the reviewer. We used PAC to match the model’s name. PAC was replaced by PDAC in the whole manuscript and the figures.

5) When describing the cohort it would be very very important to note that the training set and the test set was acquired from different institutions. Is this the case? If so, please state it explicitly. If this is not the case, you have a problem because the training set is contaminated with samples from the test set centers.

We apologize if this was not made clear enough. There is of course no overlap between the training cohort (cases aggregate from three centers) and the validations cohorts (from a fourth institution and the TCGA). We made this clearer in the M&M section and the text (P16). It now reads: “BJN_U, BJN_M are two independent validation cohorts of patients treated at a 4th center, Beaujon University Hospital, between September 1996 and January 2014.”

6) In the background section of the introduction, the authors should cite some work on the quality criteria for deep learning studies, such as Kleppe et al Nat Rev Cancer 2021 or Shmatko et al Nat Cancer 2022.

We have added these useful references in a short paragraph on AI in oncology in the introduction (P5). It now reads: “Artificial intelligence was proven to be a valuable tool to predict molecular alterations or phenotypes from histological slides potentially unlocking advanced diagnosis for all ¹⁰⁻¹³.”

7) How did the authors perform the regression learning of continuous scores? This is not described appropriately.

We thank the reviewer for his comment and modified the M&M section accordingly (See below (P20-21)). It now reads: "PACpAInt-Comp was inspired by the WELDON algorithm: 4 scores for each tile are computed from the tile features, where each score corresponds to each of the 4 molecular components. This scoring is performed using an MLP with 128 hidden neurons followed by 4 neurons and ReLU activation. For each score dimension, we select R=100 top and bottom scores and average them, so that the model's output is a vector of size 4, corresponding to the continuous predicted values of each molecular component. The model was trained with the mean squared error as loss function, using molecular components defined by RNA sequencing at the patient level as labels."

8) The source codes should be made publicly available.

We added a link to the PACpAInt GitHub repository: <https://github.com/owkin/pacpaint> (P19)

9) The methods section should be moved before the results section.

We agree that this article structure is more common, but the Nature Communications format imposes that the methods are at the end of the manuscript.

10) The results section contains a lot of non-results (methods description and background), please move this content to the appropriate sections.

We thank the reviewer for this comment and have cleaned up the results section as much as possible. Yet, since this study is dedicated to a very wide audience (clinicians, biologists, pathologists, AI specialists), we felt it was important to keep some explanatory sentences that are essential and often deeply misunderstood by non-pathologists (issue of (multiples) blocks/slides), by non-computer scientists (how the model works and makes its prediction), by non-biologists (issue of heterogeneity). We feel these issues are major in the field and often disregarded (the morphological heterogeneity with multiple tumor blocks is almost never addressed for instance). Clear manuscripts targeted to all could be a way to raise awareness about these issues.

11) The authors restrict the analysis to "to the 50% of cases that had the clearest, unambiguous". This is definitely not good practice. If you want to train a classifier for cats vs. dogs, you cannot exclude the 50% most difficult cases from your dataset, and only include the "easiest" half. Of course, this procedure boosts the AUROC but I don't consider it scientifically appropriate. Please report the results for the whole cohort. If you want to use the cleaned cohort of just the easy samples, move this to the supplements.

We agree with the reviewer that this is not good practice to develop a model or assess its performance. The point we wanted to make with this approach was that the RNA-based dichotomic classifier and therefore the proposed labels are improper in tumors where both classical and basal components are mixed (PurIST RNA-based classifier is biased toward the basal-like component) or when a tumor contains neither of them. The RNA label will be the "average" while the AI model will pick up the heterogeneity. We believe this is a major point of this study and support the rationale for a "full tile" analysis rather than the average of the slide. The weakness of RNA classifiers for these "intermediate" samples was very well

demonstrated by Topham *et al.*³. They demonstrated in their study that a fraction of cases were classified either classical or basal-like depending on the classifier used while others cases were always assigned to the same class regardless of the classifier. The latter were as expected much more homogeneous from the molecular standpoint (i.e., the equivalent of our “clear” cases). In these restricted analyses, we used this type of cases for which the RNA scoring was unambiguous (i.e., one component only).

We added in the results section that this sub-study only highlights the limitations of the RNA-based classification and that the model performance is only to be judged on the whole cohort (P7-8). It now reads: “This was particularly significant within the spatially matched validation cohort BJN_M (AUC=0.95 [0.90 - 1.0]) highlighting the limitations of the RNA-based binary classification in highly heterogeneous tumors (Fig. 2d) as previously reported rather than a true increase in the model performance that must be appreciated on the whole cohort^{14,15}.”

12) The authors need to declare adherence to STARD or TRIPOD or an equivalent guideline by the equator network.

We added a TRIPOD checklist to the submitted files and added as Suppl Table 1 the comparison of the clinical and pathological data of the test and validation cohorts.

13) I am not sure that the title is backed up by the presented results.

We understand the doubt raised by the reviewer as the manuscript describes the first AI tool to predict the molecular subtypes of pancreatic cancer based on histology. The proposed tool, PACpAInt, also enabled an original study uncovering the extensive intra-tumoral heterogeneity of pancreatic cancer. We believe that the main novelty of this study is both the development of a novel tool as well as its extensive application to investigate the state of intratumoral in these tumors. The phenotyping of more than 2000 tumor slides, with a resolution of 112 μm is as of today only feasible with histology-based AI tools. We therefore believe that the title should mainly reflect the biological and translational discoveries enabled by these types of methodologies while assuming their specific development.

14) Figure 1 has a lot of wasted white space in the top right corner, not nice.

We thank the reviewer for his suggestion and modified Figure 1 accordingly.

15) In figure 6 a and b the rainbow colors do not convey any information and should be removed (just make the forest plot black)

We agreed with the reviewer and removed colors from all forest plots (Figure 2, Supplementary Figure 2, Figure 4, Figure 6).

REVIEWERS' COMMENTS

Reviewer #2 (Remarks to the Author):

The authors have provided detailed and convincing data to address my previous comments. In my opinion, inclusion of the figure provided in the rebuttal to address point 1 in the supplemental data would strengthen the validity of the results, so I recommend the authors to do so.

Reviewer #3 (Remarks to the Author):

I thank the authors for addressing all of my comments. No further requests.

Dear Professor Cros,

In checking your recent submission to Nature Communications, we have found the following problems, which you must address before we can formally consider your manuscript.

Please ensure that all the changes requested below are addressed. If this is not done, the manuscript will be returned to you again. We may also request further changes once we have completed our checks. We require an additional point-by-point response to our editorial requests below. Please upload this using the file type 'Related Manuscript File' on our submission system, and do not modify or remove the previously uploaded cover letter or point-by-point response unless specifically asked. If you have any questions about your manuscript, please reply to this email, keeping the handling editor copied in.

*** To improve the readability of your title, we would recommend keeping the original version with punctuation: "PACpAInt: a histology-based deep learning model uncovers the extensive intratumor molecular heterogeneity of pancreatic adenocarcinoma".**

We agree and changed accordingly the title.

*** When discussing the current work in the abstract, please use the present tense. We would also recommend the following modification for the first sentence, please let us know if you agree: "Two tumor and stroma subtypes of Pancreatic adenocarcinoma (PDAC) with prognostic and theragnostic implications have been described." Please note "study the extend".**

- Because we mention later on in the abstract the basal subtype, I think we need to leave it in the first sentence so a not expert reader would know that it is a molecular subtype of tumor cell. We changed "were" for "have been" in the first sentence.

- We removed "the extend"

- We changed the tense to the present in the last sentence. We left only one sentence in the past tense "was trained" as we do not see how to put it differently. Is it ok with you?

*** Please refrain from using words such as new/novel/first or equivalent words and phrases, when referring to the scientific findings. For example: "novel Hybrid tumors" (Abstract), "the first AI-based tool", "The originality of our approach", "Its novelty lies", "never explored before", "the first clear picture" (Discussion). Please also remove exaggerated language such as "excellent"**

We made the requested changes.

*** Please provide a statement on whether informed consent was obtained by study participants in the Reporting Summary and Methods. Please also provide information on participant compensation.**

We added the following statement: Non-deceased patients were informed of the study. Patients were not compensated for their participation in the study.

*** IMPORTANT: Please note that the deposition of all raw sequencing data generated in your study in one of our approved databases is a mandatory prerequisite to accept your manuscript for publication. Please do not resubmit your revised manuscript until all the raw sequencing data are deposited and publicly available according to our policies. Please note that Zenodo is not one of our approved repositories for the deposition of RNA-seq data, and that processed count matrices are not sufficient to fulfill our data deposition requirements. Please use the link below to find a list of approved and recommended data repositories: <https://www.springernature.com/gp/authors/research-data-policy/repositories-bio/12327160>**

Raw data were deposited in array express (E-MTAB-13007). This was added in the manuscript and the data statement.

*** Thank you for depositing your code in Zenodo. Please include the Zenodo DOI in your References and cite it accordingly in your Code Availability statement; in your reference list, please include: authors, title (this paper), repository name, DOI identifier, year.**

Done

*** In your Competing Interests statement, please indicate which authors fall under "Persons affiliated with Owkin own stocks in the company."**

Done

*** Please indicate the boxplot elements in the legend of Supplementary Figure 4.**

Done

*** Please rename Extended Tables to Supplementary Tables. Please place your "TRIPOD Checklist" as a Supplementary Table or a numbered Supplementary Note.**

Done

*** Please supply a Source Data file with spreadsheets or files for all data presented in graphs within the**

Figures. Please note that this does not have to correspond to the thousands or millions of predicted tiles, but rather to the concrete data points that are shown throughout the plots in your main and supplementary figures. Within the Source Data file, the relevant raw data from each figure or table (in the main manuscript and in the Supplementary Information) should be represented by a single sheet in an Excel document, or a single .txt file or other file type in a zipped folder. An example of the Source Data file is available demonstrating the correct format:

<https://www.nature.com/documents/ncomms-example-source-data.xlsx>

The file should be labelled 'Source Data', with the title and a brief description included in your response here and should be mentioned in all relevant figure legends using the template text below: 'Source data are provided as a Source Data file.'

Done

A reference to the source data file should be added in the 'Data Availability' section, using the text "Source data are provided with this paper."

Done

* Please update and upload a final version of the Editorial Policy Checklist with your revised manuscript files. A blank Editorial Policy Checklist can be found via the link below. Note that this form is a dynamic 'smart pdf' and must be downloaded and completed in Adobe Reader.

Please update your current checklist or download from:

<https://www.nature.com/documents/nr-editorial-policy-checklist.zip>

Done

* Please provide complete name of each author in author list in article file.

Done

* Please make the affiliation number as Superscript form while tagging with the author names.

Done

* Please provide department name too in affiliations 1 and 6

Done

* In Author Contribution section, please ensure that co-author Armelle Bardier is mentioned with correct initials. Author with initials ABD do not match with any author listed. Please check.

Done

* Please rename "Disclosures" as "Competing interests policy" in article file.

Done

* Please provide each article figure separately on submission system.

Done

* Please remove Supplementary figure legends from the article file. Instead, please include them under each corresponding Supplementary Figure in your Supplementary Information file.

Done. To keep the supplementary figures large enough, we grouped the legends in a supplemental file that will be attached to the supplemental material.

Your paper has been placed back in the 'Author Approval Folder', which you may access via the following link:

<https://mts-ncomms.nature.com/cgi-bin/main.plex?el=A4S7CqNf6C6HqmS3F7A9ftdEj1MbXRY3buXQwRQwMPgZ>

Please remedy the point(s) specified above and resubmit your paper following the same steps as before.

If you have any questions, please do not hesitate to contact us.

Best regards,

Shrejal
Editorial
Nature

Bhotmange
Assistant
Communications